

# Amplification of black carbon light absorption induced by atmospheric aging: temporal variation at seasonal and diel scales in urban Guangzhou

Jia Yin Sun[1,2], Cheng Wu[1,2*], Dui Wu[1,2,3*], Chunlei Cheng[1,2], Mei Li[1,2], Lei Li[1,2], Tao Deng[3], Jian Zhen Yu[4,5,6], Yong Jie Li[7], Qiani Zhou[1,2], Yue Liang[1,2], Tianlin Sun[1,2], Lang Song[1,2], Peng Cheng[1,2], Wenda Yang[1,2], Chenglei Pei[8,9,10], Yanning Chen[10], Yanxiang Cen[11], Huiqing Nian[11], Zhen Zhou[1,2*]

[1]Institute of Mass Spectrometry and Atmospheric Environment, Jinan University, Guangzhou 510632, China

[2]Guangdong Provincial Engineering Research Center for on-line source apportionment system of air pollution, Guangzhou 510632, China

[3]Institute of Tropical and Marine Meteorology, CMA, Guangzhou 510080, China

[4]Department of Chemistry, Hong Kong University of Science & Technology, Hong Kong, China

[5]Division of Environment & Sustainability, Hong Kong University of Science & Technology, Hong Kong, China

[6]Atmospheric Research Center, HKUST Fok Ying Tung Research Institute, Guangzhou 511400, China

[7]Faculty of Science and Technology, University of Macau, China

[8]State Key Laboratory of Organic Geochemistry and Guangdong Key Laboratory of Environmental Protection and Resources Utilization, Guangzhou Institute of Geochemistry, Chinese Academy of Sciences, Guangzhou 510640, China

[9]University of Chinese Academy of Sciences, Beijing 100049, China

[10]Guangzhou Environmental Monitoring Center, Guangzhou 510030, China

[11]Guangzhou Hexin Analytical Instrument Limited Company, Guangzhou 510530, China

*Correspondence to*: Cheng Wu (wucheng.vip@foxmail.com), Dui Wu (wudui.vip@foxmail.com) and Zhen Zhou (zhouzhen@gig.ac.cn)





**Abstract.** Black carbon (BC) is an important climate forcer in the atmosphere. Amplification of light absorption can occur by coatings on BC aerosols, an effect that remains one of the major sources of uncertainties for accessing the radiative forcing of BC. In this study, the absorption enhancement factor ($E_{abs}$) was quantified by the minimum R squared (MRS) method using elemental carbon (EC) as the tracer. Two field campaigns were conducted in urban

Guangzhou at the Jinan university super site during both wet season (July 31–September 10, 2017) and dry season (November 15, 2017–January 15, 2018) to explore the temporal dynamics of BC optical properties. The average concentration of EC was $1.94\pm0.93$ and $2.81\pm2.01\mu gC$ $m^{-3}$ in the wet and dry seasons, respectively. Mass absorption efficiency at 520 nm by primary aerosols ($MAE_{p520}$) determined by MRS exhibit a strong seasonality ($8.6$ $m^2g^{-1}$ in the wet season and $16.8$ $m^2g^{-1}$ in the dry season). $E_{abs520}$ was higher in the wet season ($1.51\pm0.50$) and lower in the dry

season ($1.29\pm0.28$). Absorption Ångström exponent ($AAE_{470-660}$) in the dry season ($1.46\pm0.12$) were higher than that in the wet season ($1.37\pm0.10$). Collective evidence showed that the active biomass burning (BB) in dry season effectively altered optical properties of BC, leading to elevated MAE, $MAE_p$ and AAE in dry season comparing to those in wet season. Diurnal $E_{abs520}$ was positively correlated with $AAE_{470-660}$ ($R^2=0.71$) and negatively correlated with the AE33 aerosol loading compensation parameter ($k$) ($R^2=0.74$) in the wet season, but these correlations were

significantly weaker in the dry season, which may be related to the impact of BB. This result suggests that lensing effect was dominating the AAE diurnal variability during the wet season. The effect of secondary processing on $E_{abs}$ diurnal dynamic were also investigated. The $E_{abs520}$ exhibit a clear dependency on secondary organic carbon to organic carbon ratio (SOC/OC). $E_{abs520}$ correlated well with nitrate, implying that gas-particle partitioning of semi-volatile compounds may potentially play an important role in steering the diurnal fluctuation of $E_{abs520}$. In dry season, the diurnal

variability of $E_{abs520}$ was associated with photochemical aging as evidenced by the good correlation ($R^2=0.69$) between oxidant concentrations ($O_x=O_3+NO_2$) and $E_{abs520}$.



## 1. Introduction

Atmospheric aerosols have received great attention in recent years due to their global climatic effects and environmental effects (Anderson et al., 2003). Carbonaceous aerosols account for a large fraction of the global

aerosol mass as the main light-absorbing materials in aerosols (Kanakidou et al., 2005; Bond and Bergstrom, 2006). Black carbon (BC), which originated from incomplete combustion of hydrocarbon fuels (Johansson et al., 2018), is the dominating fraction of light absorbing carbonaceous aerosols. BC is not only an air pollutant that poses threat to the public health (Grahame et al., 2014; Apte et al., 2015), but also an essential climate forcer (Chung and Seinfeld, 2002). The BC burden in the atmosphere increased substantially since

Industrial Revolution, as evidenced by the ice core samples (Ruppel et al., 2014). Sediment cores from eastern China marginal seas also suggest that BC flux was strongly associated with human activities (Fang et al., 2018). The environmental impact of BC was elevated by its growing abundance in the atmosphere. BC had been regarded as the third most important climate forcer after carbon dioxide and methane (IPCC, 2013). The lifetime of BC in the atmosphere is shorter (< 1 week) than other greenhouse gases (Lund et al., 2018), but

BC still can be subjected to long-range transport and therefore produce large-scale influences (Ramanathan et al., 2007). On a global scale, BC can heat the atmospheric directly owing to its strong light absorption across the solar spectrum (Bond and Bergstrom, 2006), thus contributes to the warming effect (Bond et al., 2013). On a regional scale, radiative forcing of BC lead to glacier melting at high-altitude regions such as the Tibetan Plateau (Ming et al., 2008), causing seasonal water shortages in Asian rivers and affecting Asian

monsoons (Menon et al., 2002; Lau et al., 2006). On a local scale, BC can modify planetary boundary layer meteorology that leads to the "dome effect", and thus enhance local pollution indirectly (Ding et al., 2016; Wilcox et al., 2016). In microscale, BC was found playing a key role in the photochemical aging of soot by initiating the oxidation of OC (Li et al., 2018c). In addition, BC can indirectly affect the climate by altering cloud formation and cloud cover (Nenes et al., 2002; Koch and Del Genio, 2010; Kaufman and Koren, 2006;

Albrecht, 1989). Once deposit on ice and snow, BC can reduced the surface albedo, leading to the melting of ice and snow (Gertler et al., 2016; Kopacz et al., 2011; Flanner et al., 2007; He et al., 2018; Hansen and Nazarenko, 2004).

However, large uncertainties still exist in estimating the radioactive forcing of BC (Bond et al., 2013). The gap largely arises from the limited characterization of BC mixing state in the atmosphere (Fuller et al.,

1999; Jacobson, 2001; Nordmann et al., 2014). BC is chemically inert, but morphology transformation is unavoidable once emitted into the atmosphere. A recent study suggested that BC restructuring during aging can be divided into two steps (Pei et al., 2018). First, the void of the BC particles will be filled by the aging


induced materials. Once filled, further accumulation of organic and inorganic coating materials leads to the growth of particle size. Ma et al. (2013) reported soot restructuring during water evaporation in a laboratory

study. Such morphology transformation leads to alternation of BC optical properties, as evidenced by a number of laboratory experiments (Schnaiter et al., 2005; Zhang et al., 2008; Xue et al., 2009; Shiraiwa et al., 2010; Metcalf et al., 2013; Wei et al., 2013; Chen et al., 2015), field studies (Knox et al., 2009; Cappa et al., 2012; Lack et al., 2012b; Liu et al., 2015; Liu et al., 2017a) and numerical studies (Fuller et al., 1999; Bond et al., 2006; Liu et al., 2016a; Zhang et al., 2017; Lefevre et al., 2019). The presence of coating materials on

BC leads to the increase of mass absorption efficiency (MAE) through the lensing effect (Schwarz et al., 2008b). Besides coating thickness, the magnitude of light absorption enhancement by the lensing effect also depends on the optical properties of the coating materials. A coating of brown carbon (BrC) can further amplify the light absorption comparing to a transparent coating (Lack and Cappa, 2010). Recent studies suggested that BC mixing state diversity also affects the bulk $E_{abs}$ (Fierce et al., 2016; Matsui et al., 2018;

Cappa et al., 2019).

The total BC light absorption ($\sigma_{abs\_total}$) after aging can be segregated into primary absorption ($\sigma_{abs\_pri}$) by the BC core and the additional absorption ($\sigma_{abs\_aging}$) due to the presence of coating:

$$\sigma_{abs\_total} = \sigma_{abs\_pri} + \sigma_{abs\_aging} \tag{1}$$

The key parameter for light absorption enhancement, $E_{abs}$, can be calculated from:


$$E_{abs} = \frac{\sigma_{abs\_total}}{\sigma_{abs\_pri}} = \frac{MAE_t}{MAE_p} \tag{2}$$

where $MAE_t$ is the MAE of coated BC:

$$MAE_t = \frac{\sigma_{abs\_total}}{EC} \tag{3}$$

and $MAE_p$ represent the MAE of primary emitted BC:

$$MAE_p = \frac{\sigma_{abs\_pri}}{EC} \tag{4}$$

As a result, atmospheric aging process leads to BC $E_{abs}$ larger than 1.

Three technical approaches had been applied for $E_{abs}$ quantification as summarized in Table 1. The first approach is to use a thermal denuder (TD) upstream of the instrument that measures $\sigma_{abs}$ (e.g. PAS, photo-acoustic spectrometer). By measuring the denuded and ambient sample in rotation with a desired interval (e.g. 5 min), $\sigma_{abs\_total}$ and $\sigma_{abs\_pri}$ can be obtained to determine $E_{abs}$ following Eq.2. Particle loss in TD is

unavoidable and need to be accounted for (Burtscher et al., 2001). The advantage of TD is its ability to obtain





high-time-resolution data (Cappa et al., 2012; Lack et al., 2012b; Liu et al., 2015; Liu et al., 2017a). But TD has its own limitations. First, TD is not suitable for long-term measurements (e.g. most studies last for a few months). The selection of working temperature is sample depended and varied by sampling sites. As a result, a universal optimal TD working temperature does not exist. If the temperature is to low, the coating materials cannot be fully vaporized. On the other hand, if the temperature is too high, pyrolysis would occur (Irwin et al., 2013), leading to a biased $E_{abs}$ measurement. For example, Li et al. (2018a) explore the variability of TD temperature on $E_{abs}$ determination in Hong Kong. For a TD temperature of 50℃ to 200℃, $E_{abs}$ ranges from 1.02 to 1.20. $E_{abs}$ reaches 1.6 for a TD temperature 280℃. Third, the TD is not the ideal time machine for reversing the morphology transformation of BC. Previous studies have shown that the chain-like-aggregate morphology of nascent BC cannot be restored after thermodenuding of the coatings on the reconstructed BC core, which tends to be more compact and spherical (Bambha et al., 2013; Ghazi and Olfert, 2013). In addition, the high cost of TD-PAS system thwarts its wider applications in field studies.

The second approach for $E_{abs}$ determination is aerosol filter filtration-dissolution (AFD). AFD remove coatings on BC using water and organic solvent (Cui et al., 2016b). The advantage of AFD is that this method can be applied on historical filters archived by long-term/large-scale speciation sampling networks. It opens up a new path to retrieve the historical $E_{abs}$ from datasets with large temporal and spatial coverage. The limitation mainly arises from the AFD treatment process, which only removes the soluble part of the coating. The AFD treatment process is also labor intensive. The time resolution of $E_{abs}$ by AFD depends on the interval of filter sampling, which has a typical sampling time of 24 hr, making it difficult to study the diurnal pattern of $E_{abs}$.

The third approach is MAE method. $E_{abs}$ is quantified from the ratio of $MAE_t$ to $MAE_p$ as shown in Eq. 2. Since $MAE_t$ can be obtained from ambient measurements, the determination of $MAE_p$ is the key to this approach. One way is to adopt empirical $MAE_p$ in the literature (Cui et al., 2016a). Since the real-world $MAE_p$ could be highly diverse by different sources and varies temporally and spatially (Roden et al., 2006; Adler et al., 2010; Shen et al., 2013; McMeeking et al., 2014; Healy et al., 2015; Cheng et al., 2016; Weyant et al., 2016; Dastanpour et al., 2017; Radney et al., 2017; Conrad and Johnson, 2019), empirical $MAE_p$ at one site might not be applicable at other sites.

Another method to determine $MAE_p$ is combing $\sigma_{abs\_total}$ measurement with a single-particle soot photometer (SP2) to provide the mixing state of BC. The lag time between the incandescence signal and scattering can be used to differentiate thickly coated BC and bare BC. The intercept of linear regression between MAE (y axis) against the number fraction of aged BC ($r_{aged}$, x axis) represents $MAE_p$ (Lan et al.,





2013; Wang et al., 2014). This method only considers $E_{abs}$ dependency on the number fraction of aged particles and ignores the coating thickness of the aged particles, thus is only valid for a limited period of time when coating thickness and size distribution is relatively stable. An improved method for $MAE_p$

determination by SP2 is utilizing the rBC size distribution to calculate the $MAE_p$ by Mie model (Liu et al., 2017a; Wang et al., 2018a; Wang et al., 2018b).

A recently developed approach, Minimum R Squared method (MRS) can be applied to $MAE_p$ determination using elemental carbon (EC) as an tracer (Wu et al., 2018). MRS is a statistic approach and $MAE_p$ can be determined in a quantitative manner that minimizes the arbitrariness in $MAE_p$ estimation by

the traditional approach. As summarized in Table 1, $E_{abs}$ by MRS only requires co-located Aethalometer and time-resolved OC/EC measurements, which had been widely deployed around the globe, making MRS potentially more applicable than the TD approach for determining long-term variations of $E_{abs}$.

In this paper, $E_{abs}$ was determined by the MRS method using the measurement data in urban Guangzhou, a typical megacity in southern China. The aim of this study is to characterize the diurnal and seasonal patterns

light absorption enhancement of BC and its association with photochemical aging and BC mixing state. Abbreviations used in this paper are listed in Table A1 in the appendix.

## 2. Field measurements and data analysis methods

### 2.1 Characteristics of the observation site

As shown in Figure 1, sampling of this study was conducted at Jinan University atmospheric (JNU) super

site (113.35°E, 23.13°N, 40 meters above sea level), which located in Tianhe District, downtown Guangzhou. The site is on top of the library building and surrounded by teaching and residential areas. The campus was surrounded by three busiest road of the city (Figure S1) and traffic emission is a major source of primary emissions. Guangzhou is located in the southern China and is also the geographical center of Guangdong Province. There are limited industrial pollution sources around the sampling site, thus this site can represent

the typical urban environment in the Pearl River Delta (PRD) region.

The subtropical climate of PRD is strongly affected by two monsoon systems: South China Sea (SCS) monsoon and Northeast monsoon. April to May is the transition period of the Northeast monsoon to the SCS monsoon. June to September is the SCS monsoon-dominated period (wet season). The southern prevailing wind brings the clean and humid air masses from the vast ocean. October is the transition period of the SCS

monsoon to the Northeast monsoon. November to March is the Northeast monsoon-dominated period (dry season). The northeastern prevailing wind brings polluted air masses from the more economically-developed regions in the eastern Asia. This study included two sampling periods: July 31–September 10 2017 and


November 15 2017–January 7 2018, corresponding to wet and dry seasons, respectively.

**2.2 Light absorption measurements**

A dual–spot Aethalometer (Model AE33, Magee Scientific Company, Berkeley, CA, USA) was used for $\sigma_{abs\_total}$ determination. Aethalometer sampling was performed at a flow rate of 5 L min$^{-1}$ with a 2.5 μm cyclone inlet. A Nafion dryer was used to maintain the RH<40%. The data logging time resolution is 1 minute. AE33 reports results in the form of equivalent BC mass (eBC), which can be used to back-calculate the $\sigma_{abs\_total}$. MAE values from study by Drinovec et al. (2015) was adopted for $\sigma_{abs\_total}$ back-calculations at

different wavelengths. A multiple scattering correction factor C$_{ref}$=3.29 was used according to a recent study in this region (Qin et al., 2018). As a filter-based method, deposition of light absorbing particles on filter leads to the attenuation of filter transmittance signal, which is proportional to the BC mass concentration. However, as the particle deposition layer gradually increase, light was block at the upper particle layer before reaching the underneath particle layer, resulting the well-known artifact: loading effect. Since the lower layer particles

did not contribute to the light attenuation, the linear relationship between BC mass concentration and light attenuation signal was distorted.

The AE33 adopted the "dual spot" design to minimize the loading effect (Drinovec et al., 2015), which is an improvement of the traditional "single spot" correction (Virkkula et al., 2007). Two spots perform the sampling simultaneously. The correction can be implemented for each wavelength by the following two

equations,

$$eBC1_{raw} = eBC_{compensated} \cdot (1 - k \cdot ATN1) \qquad (5)$$

$$eBC2_{raw} = eBC_{compensated} \cdot (1 - k \cdot ATN2) \qquad (6)$$

Where $eBC1_{raw}$ and $eBC2_{raw}$ are the uncorrected eBC mass determined by the two spots. $eBC_{compensated}$ is the corrected eBC concentration to be determined. $k$ is the empirical compensation

parameter. $ATN1$ and $ATN2$ are the light attenuation measured at the two spots. The flows of the two spots were maintained at a ratio of 2:1 to achieve differential increase of ATN in a set window of time (e.g. 1 min). Since $BC1_{raw}$, $BC2_{raw}$, $ATN1$ and $ATN2$ are all known variables, $eBC_{compensated}$ and $k$ can be calculated for each measurement following Eqs. (5) & (6). As shown in Figure S2, $eBC1_{raw}$ and $eBC2_{raw}$ exhibit discontinuity once filter was moved to the next position, which implies biases induced by the loading

effect. After the dual spot correction, the discontinuity was minimized substantially (Figure S2).

It is worth noting that in the single spot correction, $k$ was a constant in each spot cycle, which means



all $eBC_{\text{compensated}}$ within the same cycle (e.g. a cycle last for several hours) have to share the same $k$. In contrast, time-resolved $k$ can be determined for individual $eBC_{\text{compensated}}$ in the dual spot correction, which is a useful indicator for the mixing state (Drinovec et al., 2017). Zero test was conducted monthly for

data quality control purpose.

The absorption Ångström exponent (AAE) can be determined by the multiwavelength measurement of AE33. AAE is a useful parameter to quantify the wavelength dependency of BC light absorption, as defined by the following equation (Moosmüller et al., 2011):

$$\frac{\sigma_{abs,\lambda 1}}{\sigma_{abs,\lambda 2}} = \left(\frac{\lambda_1}{\lambda_2}\right)^{-AAE} \tag{7}$$

where $\sigma_{abs,\lambda 1}$ and $\sigma_{abs,\lambda 2}$ are the light absorption coefficients at the wavelengths of $\lambda_1$ and $\lambda_2$. The AAE of freshly emitted soot from vehicular emissions is close to 1 (Bond and Bergstrom, 2006; You et al., 2016). An increase of AAE could occur due to the coating of either BrC or non-absorbing materials. Samples that strongly influenced by BB, which are generally rich in primary BrC, can inflate AAE larger than 2 (Reid et al., 2005; Lewis et al., 2008; McMeeking et al., 2009; Pokhrel et al., 2016). Beside BB influence, an increase

of AAE up to 1.5 due to coating of non-absorbing materials on the BC particles had also been observed in both model simulations (Lack and Langridge, 2013) and laboratory experiments (You et al., 2016).

**2.3 OC and EC measurements**

A filed carbon analyzer (Model RT-4, Sunset Laboratory Inc, Tigard, Oregon, USA) was used for OC and EC determination. The detailed sampling procedures can be found in our previous study (Wu et al., 2019), and

only a brief description is given here. The sample was collected in the first 45 minutes of each hour at a flow rate of 8 L min[-1]. The sample was analyzed in the next 15 minutes using thermo-optical analysis (Huntzicker et al., 1982). In the first stage, OC was vaporized by step-wise heating under helium (He) that provides an oxygen-free environment. In the second stage, carrier gas was shifted to oxygen (2% $O_2$ in He) to oxidize EC on the filter. The decomposition products of these two stages were converted to carbon dioxide ($CO_2$) by a

manganese dioxide ($MnO_2$) catalyst, then detected by a non-dispersive infrared absorption (NDIR) detector. The instrument blank was analyzed on a daily basis. Filter was changed every 6 days to minimized the bias due to the accumulation of refractory materials on the filter.

**2.4 Single particle mass spectrometry measurements**

In the wet season, a Single Particle Aerosol Mass Spectrometry (SPAMS, Hexin Analytical Instrument Co.,

Ltd., China) was deployed at Jinan university atmospheric super site during 11 to 18 August 2017. In dry season, SPAMS data (15 November 2017 to 27 December 2017) from Guangdong environmental monitoring



(GEM) site was used to characterize the EC-containing particles. The GEM site was located south to the JNU site (4 km). The operation principle of SPAMS had been introduced previously (Li et al., 2011), and only a brief introduction is given. The particles are introduced into the vacuum system through an 80 μm critical

orifice, and then focused into a particle beam by the aerodynamic lens. As a result, the particles are accelerated to a size-dependent terminal velocity. The flight time of a known distance (6 cm) for individual particles is then detected by two orthogonally-orientated continuous laser beams (Nd:YAG, 532 nm) for particle size determination. Sized particles are individually vaporized and ionized by a 266 nm pulsed laser (Nd:YAG, 0.6 mJ). The generated positive and negative ions are then detected by a Z-shaped bipolar time-of-flight mass

spectrometer. SPAMS data analysis was performed by the Computational Continuation Core (COCO, V3.2) toolkit based on the MATLAB software. In total, 327,453 and 2,212,688 particles with both positive and negative mass spectra were determined by the SPAMS in wet and dry season, respectively. Based on the ion marker criteria shown in Table S1, 120,351 and 595,180 EC-containing particles were identified in wet and dry season, respectively. EC-containing particles accounting for 37% and 27% of total detected particles in

wet and dry season, respectively, which is comparable with a previous SPAMS study in Guangzhou (Zhang et al., 2015). EC-containing particles were further grouped into two categories, EC-fresh and EC-aged. EC-aged particles were extracted from EC-containing particles using the ion markers with the relative peak area (RPA) threshold listed in Table S1, including -97 $[HSO_4]^-$, -62 $[HNO_3]^-$, -46$[NO_2]^-$, 43$[C_2H_3O]^+$, etc. Once EC-aged particles were defined, the remaining EC-containing particles are considered as EC-fresh particles.

Despite the limitations in chemical composition quantification that associated with the matrix effects induced by laser desorption/ionization, SPAMS is a unique technique that can provide chemical composition on a single particle level. The major advantage of single particle analysis by SPAMS enables the characterization of coating martials exclusively on soot particles (Li et al., 2018b), while bulk analytical techniques are incapable of distinguishing whether the non-EC materials are internally or externally mixed

with EC. Relative peak area (RPA), which was defined as the peak area of each marker ion divided by the peak area of total ions, has been recognized as an indicator of the relative amount of a species on a particle (Gross et al., 2000; Jeong et al., 2011; Hatch et al., 2014; Zhou et al., 2016). Therefore, RPA is used in this study for SPAMS data analysis.

### 2.5 Auxiliary measurements

$NO_2$ was determined by a chemiluminescence analyzer (Model 42iTL, Thermo Scientific), while $O_3$ was measured by UV photometric analyzer (Model 49i, Thermo Fisher Scientific, Waltham, MA, USA). Span and zero calibrations for the gas analyzers were performed automatedly on a weekly basis. Meteorological factors



were measured by a multi-parameter sensor (Model WXT 520, Vaisala, Vantaa, Finland). The planetary

boundary layer height (PBLH) measurements was conducted by a micro-pulse lidar (Sigma Space Co., USA)

at the Guangzhou Meteorological Bureau (GMB, 23.00° N, 113.32° E, elevation: 43 m). Hourly backward

trajectories for the past 72 hours were calculated using NOAA's HYSPLIT (Hybrid Single Particle

Lagrangian Integrated Trajectory, version 4) model (Draxier and Hess, 1998) for both dry wand wet seasons.

Backward trajectory cluster analysis was conducted using MeteoInfo (Wang, 2014, 2019). Fire count data

from Visible Infrared Imaging Radiometer Suite (VIIRS) on board the Suomi NPP weather satellite (Csiszar

et al., 2014) was downloaded from the NASA FIRMS website (https://firms.modaps.eosdis.nasa.gov/) to

generate the fire count map.

**2.6 $MAE_p$ estimation by MRS method**

$MAE_p$ is the key parameter in the $E_{abs}$ calculation. In this study, $MAE_p$ was determined by the newly

developed MRS method (Wu et al., 2018), using EC as a tracer. In MRS calculation, the correlation ($R^2$)

between measured EC and estimated $\sigma_{abs\_aging}$ is examined as a function of a series of hypothetical $MAE_p$

($MAE_{p\_h}$). The atmospheric aging induced additional light absorption, and $\sigma_{abs\_aging}$ can be calculated by

subtracting the absorption coefficient of primary aerosols, as shown in Eq.8 (a combination of Eqs. 1&4):

$$\sigma_{abs\_aging} = \sigma_{abs\_total} - MAE_p \times EC \tag{8}$$

The MAE at the minimum $R^2$ of the EC vs. $\sigma_{abs\_aging}$ relationship corresponds to the authentic $MAE_p$. The

detailed method evaluation of MRS can be found in our previous paper (Wu et al., 2018) . Only a brief

description on the calculation steps is provided here. EC from the Sunset carbon analyzer and $\sigma_{abs\_total}$ from

AE33 are used as input variables. During the calculation of $MAE_p$ by MRS, $MAE_{p\_h}$ is varied continuously

in a reasonable range. At each $MAE_{p\_h}$, corresponding hypothetical $\sigma_{abs\_aging}$ ($\sigma_{abs\_aging\_h}$) values are

calculated for the dataset and a correlation coefficient value ($R^2$) of EC vs. $\sigma_{abs\_aging\_h}$ (i.e.,

$R^2$(EC, $\sigma_{abs\_aging\_h}$)) is obtained. By searching the $MAE_{p\_h}$ in a desired range (e.g. from 0.1 to 50 with an

interval of 0.1), a series of $R^2$(EC, $\sigma_{abs\_aging\_h}$) values are then plotted against the $MAE_{p\_h}$ values (Figure 2).

The $\sigma_{abs\_pri}$ is the part of light absorption from primary emitted soot particles. As a result, $\sigma_{abs\_pri}$ is

well correlated with EC mass. In contrast, $\sigma_{abs\_aging}$ is the part of light absorption gained during the aging

processes after emission. The variability of $\sigma_{abs\_aging}$ mainly depends on the coating thickness of the soot

particles. Consequently, $\sigma_{abs\_aging}$ is independent of EC mass and the $MAE_{p\_h}$ corresponding to the





minimum $R^2$(EC, $\sigma_{abs\_aging\_h}$) would then represent the authentic MAE$_p$.

It is worth noting that MAE$_p$ by MRS represents the MAE$_p$ at the emission source, which is conceptually different from the MAE$_p$ by the TD method. First, the morphology and optical properties of freshly emitted BC particles (chain-like aggregates) is different from that of thermally denuded BC particles (compact aggregates). Second, most of the coatings are removed for TD denuded BC particles, but freshly emitted BC particles usually come with a thin coating of OC formed from condensation of organic vapors due to the temperature gradient from the flame to the ambient air. As a result, the MRS-derived MAE$_p$ is expected to be higher than the MAE$_p$ by the TD method.

**2.7 Secondary organic carbon (SOC) estimation by MRS method**

OC can be separated into two categories based on the formation nature. Primary organic carbon (POC) can be emitted from traffic emission (Huang et al., 2014), biomass burning (Simoneit, 2002), trash burning and cooking (Mohr et al., 2009). Secondary organic carbon (SOC) can be formed through oxidation of volatile organic compounds (VOCs) or semi-volatile POC (Hallquist et al., 2009). The EC tracer method had been used extensively for SOC estimation (Turpin and Huntzicker, 1995):

$$POC = (OC/EC)_{pri} \times EC + OC_{non-comb} \qquad (9)$$

$$SOC = OC_{total} - POC \qquad (10)$$

Combing Eqs. (9)&(10):

$$SOC = OC_{total} - (OC/EC)_{pri} \times EC - OC_{non-comb} \qquad (11)$$

(OC/EC)$_{pri}$ represents the overall OC/EC ratio of aerosols from primary emission sources, while OC$_{non-comb}$ represents primary OC from non-combustion process. OC$_{non-comb}$ can be determined from the intercept of OC vs. EC linear regression. In this study, weighted orthogonal distance regression (WODR) was used to account for errors in both x and y variables (Wu and Yu, 2018). By grouping the data into percentile subsets using OC/EC ratio from the lowest to the highest (1–100%, with an interval of 1%), a series of intercepts were obtained as a function of OC/EC percentile (Figure S3). The intercept term in the OC vs. EC WODR is very small (-0.88 – -0.05) throughout the percentile range (1–100%). Since this term is small, OC$_{non-comb}$ was set to zero for SOC estimation in this study.

(OC/EC)$_{pri}$ is the key parameter for SOC calculation in the EC tracer method. In MRS method, the correlation ($R^2$) between measured EC and estimated SOC (from Eq. 10) was examined as a function of a series of hypothetical (OC/EC)$_{pri}$ ((OC/EC)$_{pri\_h}$). The OC/EC ratio at the minimum $R^2$ (EC vs. SOC)



corresponds to the authentic primary OC/EC ratio (Millet et al., 2005). The detailed calculation steps can be found in our previous paper (Wu and Yu, 2016). Only a brief description is given here. In MRS calculation, $(OC/EC)_{pri\_h}$ was varied continuously in a reasonable range (e.g. from 0.1 to 10 with an interval of 0.1). Hypothetical SOC ($SOC_h$) values were calculated at individual $(OC/EC)_{pri\_h}$ for the whole dataset. A series of $R^2$ values of EC vs. $SOC_h$ (i.e., $R^2(EC,SOC_h)$) were generated and then plotted against the $(OC/EC)_{pri\_h}$ values. Based on the assumption that variations of EC and SOC are independent, the $(OC/EC)_{pri\_h}$ corresponding to the minimum $R^2(EC,SOC_h)$ would then indicate the authentic $(OC/EC)_{pri}$ ratio.

In our previous work, numerical studies were performed and the results showed that the minimum R squared method (MRS) is more robust in SOC estimation than the minimum OC/EC and percentile OC/EC method (Wu and Yu, 2016). As a result, the MRS method had been gradually adopted for SOC estimation in recent studies (Xu et al., 2018b; Bian et al., 2018; Ji et al., 2018; Ying et al., 2018; Ji et al., 2019; Wu et al., 2019).

An Igor Pro (WaveMet rics, Inc. Lake Oswego, OR, USA) based computer program (Wu and Yu, 2016) was used to implement MRS calculation. Another two Igor Pro-based computer programs, Histbox (Wu et al., 2018) and Scatter Plot (Wu and Yu, 2018), were used for generating the box plots and scatter plots presented in this study. These computer programs (with operation manuals) can be downloaded freely from https://sites.google.com/site/wuchengust.

## 3. Results and discussions

### 3.1 Seasonality of carbonaceous aerosols concentrations and optical properties

The time series of EC, OC, optical properties and supporting measurements during the wet and dry seasons are shown in Figure S4. The hourly EC concentrations ranged from 0.43 to 7.40 and 0.54 to 12.04 µgC m⁻³ in wet and dry seasons, respectively. As for OC, the hourly average ranged from 0.32 to 13.84 and 0.51 to 25.31 µgC m⁻³ in wet and dry seasons, respectively. The hourly OC/EC ratios ranged from 0.25 to 6.92 and 0.33 to 8.69 in wet and dry seasons, respectively. In wet season, the wind direction is southeasterly dominated, bringing the relatively clean background air masses from the vast ocean. In dry season, the northeasterly wind prevails, which promotes the long-range transport of air pollutants from the east and central China.

As shown in Figure 1, EC, OC and OC/EC AAE₄₇₀₋₆₆₀ all exhibit clear seasonality. Average EC concentrations (with 1 standard deviation, hereafter) were 1.94±0.93 and 2.81±2.01 µgC m⁻³ in wet and dry seasons, respectively. The EC level was comparable to the measurements made in 2012 at a Guangzhou suburban site (1.67±1.35 µgC m⁻³ in wet season, 3.47±2.75 µgC m⁻³ in dry season) (Wu et al., 2019). Back trajectories analysis showed that in wet season most air masses were from South China Sea, with only 31.67%





air masses were locally influenced (Figure S5a). Among the 3 back trajectory clusters in wet season, the locally influenced air masses exhibit the shortest back trajectory (C1), leading to the highest EC concentration ($2.01\pm1.22$ µgC m$^{-3}$). As the back trajectory distance increased from C2 to C3, the corresponding EC decreased from $2.01\pm 0.88$ to $1.56 \pm 0.67$ µgC m$^{-3}$. During dry season, air masses were dominated by those of northern origin. EC concentrations also exhibit gradient on the trajectory path length. Long-path (C3) trajectories lead to low EC concentrations ($2.01\pm1.46$ µgC m$^{-3}$), while short-path trajectories lead to higher concentrations (C1: $3.50\pm2.13$ µgC m$^{-3}$; C2: $2.52\pm1.91$ µgC m$^{-3}$).

The average concentrations of OC doubled in dry season ($7.02\pm5.19$ µgC m$^{-3}$) comparing to those in wet season ($3.38\pm1.93$ µgC m$^{-3}$), leading to elevated OC/EC ratio in dry season ($2.56\pm0.94$) in contrast to wet season ($1.78\pm0.83$). The hourly AAE$_{470-660}$ ranged from 1.14 to 1.67 and 1.07 to 1.76 in wet and dry seasons, respectively. As shown Figure S6a, AAE$_{470-660}$ observed in dry season ($1.46\pm0.12$) was significantly (P<0.001) higher than that in wet season ($1.37\pm0.10$).

Measured MAE$_{520}$ in dry season ($18.47\pm5.49$ m$^2$ g$^{-1}$) is significantly (P<0.001) higher than that in wet season ($10.73\pm4.96$ m$^2$ g$^{-1}$), as shown in Figure S6b. The elevated MAE during dry season was likely a result of BB influences, which will be discussed in detail in section 3.2. The dependence of MAE$_{520}$ on wind speed and wind direction was investigated in Figure S7, which is generated using ZeFir (Petit et al., 2017). In wet season, the southeast wind dominates (Figure S7b), which is in consistency with back-trajectory analysis discussed above. MAE$_{520}$ did not show obvious dependence on wind speed and wind direction in the wet season (Figure S7a). In dry season, the northwestern wind prevailed (Figure S7d). High MAE$_{520}$ was spotted from west with a wind speed at 7 m s$^{-1}$ (Figure s7c), suggesting regional transport of aged air masses.

The MAE$_{p520}$ values determined by MRS were 8.6 and 16.8 m$^2$g$^{-1}$, for wet and dry seasons, respectively (Figure 2a & b). Similar to MAE$_{520}$, the increase of MAE$_{p520}$ in the dry season was also likely a result of BB influence, which could lead to larger BC cores (Ditas et al., 2018) and with thicker primary coatings (Schwarz et al., 2008a; Kondo et al., 2011; Lack et al., 2012a; Liu et al., 2014). More details of the BB influences will be discussed in section 3.2. Consequently, light absorption enhancement was found to be more pronounced in wet season (E$_{abs520}$=$1.51\pm0.50$, Table 2) than in dry season ($1.29\pm0.28$), because E$_{abs}$ depends on the ratio of MAE to MAE$_p$, not their absolute values. The E$_{abs}$ determined around the world shown diverse results (Table 2). Low E$_{abs}$ were found in California (1.06@532 nm)(Cappa et al., 2012) and Japan (1.06@532 nm) (Ueda et al., 2016). Liu et al. (2017a) observed a moderate E$_{abs}$ in UK (1.0-1.3@532 nm) and suggested that the small E$_{abs}$ observed by Cappa et al. (2012) was a result of mixing state diversity. A recent study in California (Cappa et al., 2019) found moderate E$_{abs}$ at Fresno (1.22@532 nm) but low E$_{abs}$ at Fontana



(1.07@532 nm), which was partially associated with unequal distribution of coating between different BC-containing particle types (Lee et al., 2019). In general, higher $E_{abs}$ values had been observed in more polluted urban areas, such as France (Paris, 1.53@880 nm) (Zhang et al., 2018a), India (Kanpur, 1.8@781 nm)(Thamban et al., 2017) and various locations in China (Wang et al., 2014; Xu et al., 2016; Zhang et al., 2018b; Lan et al., 2013; Wu et al., 2018; Cui et al., 2016b; Chen et al., 2017; Bai et al., 2018; Xie et al., 2019).

Dependence of $E_{abs520}$ on air masses was investigated by back-trajectory cluster analysis as shown in Figure S5. In wet season, the highest $E_{abs520}$ (1.71± 0.58) was found from the shortest back trajectories (C1), suggesting the local episodic events. The high $E_{abs520}$ from C1 (northeasterly air mass) was also confirmed by the wind rose plot (Figure S8). The elevated $E_{abs520}$ of C1 was likely associated with high SOC/OC ratios from Aug 18 to 23 as shown by the time series plot in Figure S4a.The two oceanic air mass clusters (C2 and C3) exhibit deviated $E_{abs520}$ characteristics. C3 represents more aged oceanic airs masses as evidenced by the lower EC (1.56 ± 0.67) and higher $E_{abs520}$ (1.58 ± 0.57). In contrast, C2 has relatively lower $E_{abs520}$ (1.40 ± 0.38) but higher EC. In dry season, $E_{abs520}$ did not show clear dependence on back trajectory clusters as the $E_{abs520}$ falls into a narrow range (1.24-1.34) between C1-C3. $E_{abs}$ dependence on wind speed and wind direction was examined in Figure S8. $E_{abs520}$ showed little dependence on wind speed and the high $E_{abs}$ occurrences (Figure S8) was largely overlapped with the high MAE as shown in Figure S7.

In summary, as evidenced by $AAE_{470-660}$ and MAE results, carbonaceous aerosols exhibit a strong seasonality in urban Guanghzou. This seasonality was associated with two seasonal factors, including contrasted direction of the prevailing wind, and diverse primary BC optical properties induced by seasonal BB influence.

### 3.2 Influence of biomass burning on the BC optical properties during dry season

Evidence from particle chemical compositions showed that BB influence was more intense in the dry season. Levoglucosan had been widely accepted as the tracer for BB in the $PM_{2.5}$ (Engling et al., 2006; Bhattarai et al., 2019). As shown in Figure S9, levoglucosan concentrations in Guangzhou were elevated by one order of magnitude during the dry season (159.33 ng m$^{-3}$) comparing to those in the wet season (35.93 ng m$^{-3}$). Besides levoglucosan, primary OC/EC ratio can also be used as an indicator of BB influence since the BB influence samples has a higher OC/EC ratio than that from traffic emissions (Schmidl et al., 2008; Pokhrel et al., 2016). In this study, $(OC/EC)_{pri}$ determined by MRS in dry season (2.31) was higher than that in wet season (1.49), as shown in Figure S10. In addition, the northeasterly wind prevailed during the dry season, which favors long-range transport of aerosols from BB from central and eastern China to the PRD region. Remote sensing



results also confirmed the more intense BB in dry season, as shown by gridded fire count map (Figure S11) determined by VIIRS.

As a result, the optical properties of BC were largely affected by the intense BB influences during the
dry season. First, as shown in Figure S6b, significantly (P<0.001) higher $MAE_{520}$ was observed in dry season (18.47±5.49) comparing to that in wet season (11.28±9.88). A previous field study at a suburban site in Guanghzou also reported the influence of BB on MAE, which observed a positive correlation between $K^+$ and MAE (Wu et al., 2018). High MAE from BB had been reported in BB emission studies as well (Roden et al., 2006; Schmidl et al., 2008; Levin et al., 2010; Wang et al., 2018a). Single particle soot photometer
(SP2) studies have shown that BB influenced BC particles are more likely to have larger BC cores (Ditas et al., 2018), and with thicker initial coatings than those from vehicular emissions (Schwarz et al., 2008a; Kondo et al., 2011; Lack et al., 2012a; Liu et al., 2014). This is in good agreement with the $MAE_{p520}$ obtained in the present study, which was almost doubled in dry season (16.8 $m^2g^{-1}$) comparing to that in wet season (8.6 $m^2g^{-1}$).

In dry season the $E_{abs}$ showed little wavelength dependence (Figure 2d) despite the influence from BB. In this sense, the BB influence did substantially alter the optical properties of primary BC in the dry season, but the contribution of secondary BrC on $E_{abs}$ was likely limited. The weak wavelength dependence of $E_{abs}$ was also observed a previous study at a suburban site in Guanghzou (Wu et al., 2018). A previous study in Guangzhou also found that the seasonal difference of BrC light absorption contribution at 405 nm between
dry season (15-19%) and wet seasons (12-15%) was small (Li et al., 2018d). In addition, the small seasonal difference of AAE between wet (1.37±0.10) and dry (1.46±0.12) seasons observed in this study also implies that secondary BrC contribution was not the dominating driver for AAE deviation from 1, which was the typical AAE for fresh soot without atmospheric aging. The results found in PRD were in contrast to a study in Paris, which found systematic higher $E_{abs370}$ than $E_{abs880}$ at wintertime due to the influence of biomass
burning (Zhang et al., 2018a).This discrepancy implies the complex linkage between BB and BrC optical properties.

The complex relationship between AAE and BrC can be affected by a variety of factors. First, the optical properties of primary BrC from BB exhibit large diversity in previous studies (Martinsson et al., 2015; Tian et al., 2019a) , which can be affected by fuel type and combustion conditions (Reid et al., 2005; Roden et al.,
2006). Second, atmospheric aging can lead to AAE elevation through the formation of secondary BrC from a variety of pathways (Moise et al., 2015; Laskin et al., 2015), including nitration of aromatic compounds (Jacobson, 1999), reaction of ammonia (Bones et al., 2010), bond-forming reactions between SOA



constituents (Shapiro et al., 2009), reactions of biomass burning products (Gilardoni et al., 2016; Kumar et al., 2018), photo-enhancement (Hems and Abbatt, 2018; Liu et al., 2016b; Ye et al., 2019), and aqueous-phase

reactions (Lin et al., 2015; Tang et al., 2016; Xu et al., 2018a).On the other hand, AAE decrease could also occur during atmospheric aging (Romonosky et al., 2019), either induced by photo-bleaching of BrC (Adler et al., 2011; Zhong and Jang, 2011, 2014; Lee et al., 2014; Canonaco et al., 2015; Lin et al., 2016; Sumlin et al., 2017; Bhattarai et al., 2018; Fortenberry et al., 2018; Hems and Abbatt, 2018; Browne et al., 2019; Dasari et al., 2019; Li et al., 2019a; Wong et al., 2019), or aqueous-phase BrC degradation in the absence of light

(Santos and Duarte, 2015; Santos et al., 2016b; Santos et al., 2016a; Fan et al., 2019). The relative contribution of secondary BrC formation and BrC degradation on the total BrC light absorption budget is still poorly understood. BrC degradation could be one of the reasons of the small seasonal AAE difference observed in the PRD region. More studies are needed by incorporating both time-resolved optical measurements and time-resolved detailed chemical speciation measurements to better understand the balance of BrC formation

and degradation.

**3.3 Diurnal dynamics of carbonaceous aerosols concentrations and optical properties**

The diurnal variations of EC, OC, OC/EC, SOC, $AAE_{470-660}$ and $E_{abs520}$ in wet and dry seasons are shown in Figure 3. Two peaks can be observed for EC (Figure 3a), one in the early morning (7:00) and the other in the evening (19:00), which reflects local traffic emissions in two rush hours. The lowest EC was found in the

afternoon (14:00), likely associated with two factors considering the nature of EC source exclusive from primary emissions. The first factor is planetary boundary layer (PBL) height. As shown in Figure S12, the diurnal maximum PBL height was at 14:00 and 15:00 for wet and dry seasons, respectively. The fully developed PBL would help dilute the concentrations of primary pollutants (Deng et al., 2016; Liu et al., 2019a; Williams et al., 2019). The second factor is the diurnal variations of traffic volume. Previous studies (Yao et

al., 2013; Xie et al., 2003) showed that traffic volume during 12:00 – 15:00 is lower than those in the morning and evening rush hours. The combination of these two factors leads to the reduced EC concentrations in the afternoon. The diurnal pattern of EC is similar between wet and dry seasons, but the magnitude was greatly elevated in the dry season.

OC exhibits a bimodal distribution (Figure 3b), peaking at 13:00 and 19:00, respectively. OC can be both

primary and secondary, making its diurnal pattern different from that of EC. OC/EC also has two peaks as shown in Figure 3c. The first peak appeared at 13:00 and the second peak showed up at 17:00. It is worth noting that in wet season the afternoon OC/EC peak was higher than that in the evening OC/EC peak, while in dry season the reverse is true.


As shown in Figure 3d, two SOC peaks are observed in wet season, with the first SOC peak at 13:00 and the

second SOC peak at 19:00. While in dry season the afternoon SOC peak was merged into the broadened

evening peak. Despite the higher SOC concentrations observed in dry season, SOC formation was more active

during the wet season as evidenced by the diurnal SOC/OC ratios (Figure S13a). The diurnal SOC/OC in wet

season was always higher than that in dry season. It is worth noting that in wet season, despite that the SOC

evening peak was comparable to the afternoon peak as shown in Figure 3d, the SOC/OC evening peak was

smaller than the afternoon peak (Figure S13a). This observation implies that the SOC evening peak in the

wet season was a result of the combination of pollutant accumulation (e.g. PBL decrease after sunset) and

SOC formation, rather than the formation process alone. The small evening peak of SOC/OC in summer

(19:00-21:00) would also be likely a result of condensation of semi-volatile organic compound (Warren et

al., 2009; Pathak et al., 2008; Liang et al., 1997) due to the temperature decrease after sunset.

SOC/OC dependence on RH was investigated (Figure S13) to explore the effect of aqueous-phase

secondary organic aerosol formation. During wet season, SOC/OC decreases as RH increases and the results

were the same for both daytime and nighttime (Figure S13a&b). During nighttime when no solar radiation

was supplied, higher RH leads to a lower SOC/OC (Figure S13b). This piece of evidence suggests that

aqueous-phase reactions were unlikely the dominating pathway for SOC formation during the wet season. In

dry season, SOC/OC did not show clear dependence on RH, suggesting that SOC formation is not sensitive

to RH in dry season.

The diurnal trend of AAE$_{470-660}$ was similar between wet and dry seasons, which is higher in the evening

and lower during the midday, but the magnitude of AAE$_{470-660}$ slightly increased during the dry season. The

E$_{abs520}$ exhibit different diurnal patterns between the wet and dry seasons. As shown in Figure 4, elevated

E$_{abs520}$ was found during nighttime in wet season but in dry season inflated E$_{abs520}$ was observed in the

afternoon. In addition, the degree of light absorption enhancement was more pronounced in the wet season.

The influencing factors of dynamics are discussed in the following sections.

### 3.4 The diurnal correlations between AAE, $k$ and E$_{abs}$

The loading effect correction factor used in AE33, $k$, has been found to be a useful indicator for the light

absorption enhancement of BC (Drinovec et al., 2017). As shown in Figure 4a, in wet season a good anti-

correlation was found between $k_3$ ($k$ value for 520 nm) and E$_{abs520}$ with a $R^2$ of 0.74. In dry season, such anti-

correlation was substantially weakened (R$^2$ = 0.20) as shown in Figure 4b, likely due to the influence of BB.

These results agree well with the findings reported by Drinovec et al. (2017) that $k$ can be used as a BC

particle coating indicator without the influence of BB. As shown in Figure 4a, a good correlation was found





between $AAE_{470-660}$ with $E_{abs520}$ ($R^2$=0.71). Considering the weak BB influence in wet season as discussed in section 3.2, atmospheric aging induced coating on BC particles was more likely the dominating driver of $AAE_{470-660}$ dynamics during the wet season in the PRD region. The presence of coating of BC could also explained that despite the BB influence is small in wet season, the observed average AAE (1.37±0.10) was significantly higher than the AAE of fresh BC (~1). This result is also consistent with previous studies that

found non-light-absorbing coating can lead to elevated AAE up to 1.5 (Lack and Cappa, 2010; Lack and Langridge, 2013). In dry season, the variability of AAE was governed by both coating thickness and BB influence, thus leading to a degraded $R^2$ (0.22) between $AAE_{470-660}$ and $E_{abs520}$ as shown in Figure 4b. A recent study showed that the diurnal pattern of BrC was moderately correlated with a BB tracer $K^+$ in the PRD region during the dry season (Li et al., 2019b), implying that BB did have considerable influence on AAE

viability during dry season.

   The spectral fingerprints of $k$ were shown in Figure S14. Observations in Europe showed that the presence of BrC could lead to increased $k$ at longer wavelengths (Drinovec et al., 2017). Our observations showed that the seasonal difference in spectral fingerprints of $k$ between wet and dry seasons is small. Considering the limited increase of AAE in dry season and similarity of seasonal spectral fingerprints of $k$,

these results suggest that, in the PRD region, despite the BB influence in dry season effectively altered the optical properties of BC aerosols, there was likely limited secondary BrC contribution on $E_{abs}$ during the dry season, which is in agreement with discussions in section 3.2.

**3.5 The influence of secondary processing on $E_{abs}$ diurnal dynamic**

Photochemical reactions play an important role in the aging process of black carbon, leading to the

modification of BC morphology and optical properties as evidenced by laboratory studies (Saathoff et al., 2003; Schnaiter et al., 2005; Martinsson et al., 2015; Pei et al., 2018) and quasi-atmospheric chamber studies (Peng et al., 2017; Peng et al., 2016). Filed studies at various locations have also showed that photochemical processing can promote the light absorption enhancement of BC, including in Beijing (Liu et al., 2019b), Yangtze River Delta (Xu et al., 2018c), Xi'an (Wang et al., 2017c), Los Angels (Krasowsky et al., 2016) and

Toronto (Knox et al., 2009). The concentration of odd oxygen ($O_x = O_3 + NO_2$) proposed by Liu (1977) and (Levy II et al., 1985) have been widely used as the indicator of photochemical aging. In this study, the diurnal correlations between $O_x$ and $E_{abs520}$ were investigated to explore the effect of photochemical processing. As shown in Figure 5a, in wet season $O_x$ and $E_{abs520}$ peaked at 15:00 and 0:00, respectively. The $O_x$ experienced a continuous decline since 15:00 until the sunrise of the next day, but the growth of $E_{abs}$ extended to midnight.

The nighttime $E_{abs520}$ peak suggests that the increase of coating can be achieved without the presence of solar





radiation. These differences in the diurnal patterns led to a low correlation between $O_x$ and $E_{abs}$ ($R^2$=0.01). This result implies that in wet season the diurnal variability of $E_{abs520}$ was unlikely dominated by photochemical reactions, despite that $O_x$ was more pronounced in the wet season. As for dry season (Figure 5b), both $O_x$ and $E_{abs520}$ peaked at 17:00 leading to a good correlation with a $R^2$ of 0.69, suggesting that

photochemical reactions could be one of the main drivers for $E_{abs}$ diurnal variations. This result strongly indicates that BC light absorption can be markedly amplified through photochemical reactions. Our dry-season results are consistent with a previous study in Northern China (Wang et al., 2017c), which also showed the dependence of light absorption enhancement on $O_x$ during the wintertime.

In the meantime, the formation of SOC also contributes to light absorption enhancement of BC, which had

been observed in both field studies (Moffet and Prather, 2009; Wang et al., 2017a; Zhang et al., 2018a) and laboratory studies (Schnaiter et al., 2005; Lambe et al., 2013; Saliba et al., 2016). In this study, the effect of SOC formation on $E_{abs}$ was investigated using SOC/OC as the indicator rather than using SOC alone. The advantage of using SOC/OC is that the SOC variations induced by non-secondary-formation process (e.g. PBLH shallowing) can be minimized, thereby focusing the analysis on the effect of secondary formation

processes. A good diurnal correlation between SOC/OC and $E_{abs520}$ were observed in dry season ($R^2$=0.53), but no correlation was found in wet season ($R^2$=0.01). The $E_{abs}$ dependence on SOC/OC was examined in Figure 6. $E_{abs}$ dependence on SOC/OC was found in both wet and dry seasons, but a clearer dependence was observed in the dry season. The results from Figure 6 suggest that the poor diurnal correlation between SOC/OC and $E_{abs520}$ observed in wet season (Figure 5a) does not necessarily rule out the contribution of SOC

on $E_{abs520}$. A study in Paris (Zhang et al., 2018a) found that more oxidized oxygenated organic aerosols (MOOOA) and less oxidized OOA (LO-OOA), which are surrogates of SOA, were the dominating contributors for $E_{abs}$, especially in summertime. In the present study, due to the lack of quantitative chemical speciation data, quantification of contributions from different chemical species on $E_{abs}$ is not possible. A recent study in Guangzhou (Wu et al., 2019) found that traffic-derived SOC could be a significant source of

SOC in the urban area, which can account for half of the total SOC. In that sense, traffic emissions are expected to have a considerable contribution to BC light absorption enhancement in both wet and dry seasons.

**3.6 The influence of semi-volatile compounds on $E_{abs}$ diurnal variations**

The SPAMS data from both wet (August 11-18, 2017) and dry season (15 November 2017 to 27 December 2017) were analyzed to explore the mixing state of EC-containing particles from a single-particle perspective.

The average EC-fresh and EC-aged mass spectra are shown in Figure S15 for both wet and dry seasons. The domination of EC-aged particles in EC-containing particles number fraction suggest that most of the EC





particles are internally mixed with other species (Table S2). This result agrees with previous studies in this region (Zhang et al., 2013; Zhang et al., 2014).

To study the relative abundance of coating materials on EC particles, we investigate the ratios of RPA by different species (organics, sulfate and nitrate) to RPA by EC in both wet and dry seasons (Figure 7). In wet season, organics and sulfate on EC-containing particles demonstrated similar diurnal trends that both peaked at 13:00, implying the association with photochemical reactions. The timing of the organic peak by SPAMS shown in Figure 7a also agrees well with the bulk measurements of SOC/OC (Figure 5a). However, the diurnal variations of organics and sulfate were poorly correlated with $E_{abs520}$ as shown by the low $R^2$ in the

scatter plot in Figure 7a&b. It should be noted that poor diurnal correlations do not necessarily rule out the contribution of organics and sulfate to $E_{abs}$ by analogy with the SOC correlation with $E_{abs}$ as discussed in section 3.5. Although the quantitative contribution estimation of sulfate and SOA to $E_{abs}$ is not possible in this study, a rough estimation can be projected. Considering the typical annual average SOC concentration (3 $\mu gC\ m^{-3}$)(Wu et al., 2019), typical SOA/SOC mass ratio (1.8) (Li et al., 2017), and sulfate concentration (8

$\mu g\ m^{-3}$)(Liu et al., 2017b) in the PRD region, SOA and sulfate would likely have comparable contributions to the $E_{abs}$, according to the $E_{abs}$ dependency on sulfate-to-SOA mass ratio results by Zhang et al. (2018a). Summertime nitrate was low in daytime and high in nighttime (Figure 7c), which agrees with measurements at the roadside site in Hong Kong (Lee et al., 2015) and Shanghai (Li et al., 2018b). Temperature-dependent gas-particle partitioning would be one of the possible reasons for the observed nitrate diurnal pattern (Appel

et al., 1981; Xue et al., 2014; Griffith et al., 2015). Higher temperature during the daytime (Figure S16) favors $HNO_3$ partitioning into the gas-phase in wet season. The diurnal pattern of nitrate correlates well with that of $E_{abs520}$ ($R^2$=0.59) as shown in Figure 7c, suggesting that $E_{abs520}$ was likely affected by temperature-induced gas-particle partitioning during wet season. A previous chamber study had shown the decrease of $E_{abs}$ due to SOA evaporation (Metcalf et al., 2013). By analogy with nitrate, organic compounds with a volatility similar

to nitrate might potentially involve in shaping the diurnal pattern of $E_{abs}$ in wet season.

In dry season, organics were moderately correlated with $E_{abs520}$ ($R^2$=0.38) as shown in Figure 7d. The improved correlation of organics in dry season was in agreement with $O_x$ and SOC/OC results as shown in Figure 5b. Sulfate was still poorly correlated with $E_{abs520}$. Since the contribution of sulfate on $E_{abs520}$ cannot be ruled out, one possible explanation is that the contribution of sulfate on $E_{abs520}$ was not reflected on the

diurnal time scale.

## 5. Conclusions and implications

This study explored the temporal dynamics of optical properties of carbonaceous aerosols in urban





Guangzhou, a typical megacity in southern China, focusing on the atmospheric aging induced light absorption enhancement of BC. Field measurements were conducted at an urban site during wet season (July 31–

September 10, 2017) and dry season (November 15, 2017–January 15, 2018). A newly developed approach, the minimum R squared (MRS) method (Wu et al., 2018), was applied to determine the light absorption enhancement factor, $E_{abs}$, using data from a Aethalometer and a field-deployable semi-continuous carbon analyzer. The MRS approach avoids specialized instrument setup (e.g. thermal denuder and photo-acoustic spectrometer) for $E_{abs}$ determination, hence has a great potential for expending data pool of $E_{abs}$, considering

the fact that collocated Aethalometer and field carbon analyzer measurements have been widely deployed around the world.

A strong seasonality of BC was observed. The average concentration of EC was $1.94\pm0.93$ and $2.81\pm2.01$ $\mu gC\ m^{-3}$ in the wet and dry seasons, respectively. Collective evidence from remote sensing fire counts and ground measurements of levoglucosan showed that biomass burning (BB) was more active in the dry season.

Consequently, optical properties of BC were effectively altered, leading to elevated MAE (dry season: $18.47\pm5.49\ m^2\ g^{-1}$, wet season: $10.73\pm4.96\ m^2\ g^{-1}$), $MAE_p$ (dry season: $15.8\ m^2\ g^{-1}$, wet season: $8.1\ m^2\ g^{-1}$) and AAE (dry season: $1.46\pm0.12$, wet season: $1.37\pm0.10$) in dry season comparing to those in wet season. However, little dependence of $E_{abs}$ on wavelength was observed in dry season despite the influence from BB. The diurnal correlation analysis between AAE, $k$ and $E_{abs}$ revealed different results between wet and dry

seasons. During the wet season when BB influence was small, AAE was well correlated with $E_{abs}$, implying that coating was likely the main driver for AAE>1. In other words, the two component AAE model might not be suitable for BrC absorption estimation under such circumstance. The aethalometer loading effect correction factor, $k$, was confirmed to be a useful $E_{abs}$ indicator owing to its good correlation with $E_{abs}$ during the wet season. In dry season, the weak correlation between AAE and $E_{abs}$ implies the contribution from BB

on AAE. In dry season, the BB influence leads to poor correlation between $k$ and $E_{abs}$, confirming that $k$ can only be used as the coating indicator when BB influence is small.

The effect of atmospheric aging on $E_{abs}$ diurnal pattern was examined. $O_x$ and SOC/OC were found well correlated with $E_{abs}$ during the dry season but no correlation was observed in the wet season. However, further analysis showed $E_{abs}$ dependence on SOC/OC in both wet and dry season. This observation implies that poor

diurnal correlation in wet season does not necessarily rule out the $E_{abs}$ contribution from SOC. In other words, the SOC contribution on $E_{abs}$ in wet season was not necessarily be reflected in mere diurnal correlation. A good diurnal correlation between nitrate and $E_{abs}$ was observed, implying the potential role of semi-volatile components in regulating the diurnal dynamics of $E_{abs}$. In China, the sulfate problem had been effectively





mitigated by the reduction measures implemented in recent years (Xia et al., 2016; Wang et al., 2017b). In

contrast, nitrate increased substantially in the recent years (Xu et al., 2019; Tian et al., 2019b). As a result,

the increasing nitrate might potentially affect BC's radiative forcing in China.



# Appendix

**Table A1. Abbreviations.**

| Abbreviation | Definition |
| --- | --- |
| $AAE_{470-660}$ | Ångström absorption exponent between 470 and 660 nm |
| AFD | aerosol filter filtration-dissolution |
| BB | biomass burning |
| BC | black carbon |
| BrC | brown carbon |
| $E_{abs520}$ | light absorption enhancement factor at 520 nm |
| $\sigma_{abs520}$ | light absorption coefficient at 520 nm |
| $\sigma_{abs\_total}$ | total light absorption coefficient of a coated particle |
| $\sigma_{abs\_pri}$ | primary light absorption coefficient attributed to the soot core alone of a coated particle |
| $\sigma_{abs\_aging}$ | extra light absorption other than $\sigma_{abs\_pri}$, including those from the lensing effect arise from non-absorbing coating on the soot core and secondary brown carbon during atmospheric aging |
| eBC | equivalent BC mass concentration determined by optical methods (e.g. aethalometer) |
| $k_1, k_2 \dots k_7$ | compensation factors (Eqs. 5 & 6) at 7 wavelengths (370,470,520,590,660, 880 and 950 nm). |
| $MAE_{520}$ | mass absorption efficiency at 520 nm, also known as mass absorption cross section (MAC) |
| $MAE_p$ | primary MAE of freshly emitted soot particles |
| $MAE_{p\_h}$ | a series of hypothetical $MAE_p$ tested in MRS calculation |
| MRS | minimum R squared method |
| OC | organic carbon |
| PRD | Pearl River Delta region, China |
| rBC | refectory black carbon (commonly used for reporting BC detected by SP2) |
| $r_{aged}$ | The ratio of aged particles to fresh particles determined by SP2 |
| SP2 | single-particle soot photometer |
| SSA | single-scattering albedo |
| TD | thermodenuder |





**Author contributions**

C.W. designed the study. J.Y.S. and C.W. performed the experiments. J.Y.S., C.W., C.C., Q.Z. and Y.L. conducted the data analysis. J.Y.S. and C.W. wrote the paper with the inputs from all authors.

**Data availability**

OC, EC, and $\sigma_{abs}$ data used in this study are available from corresponding authors upon request.

**Competing interests**

The authors declare that they have no conflict of interest.

**Acknowledgements**

This work is supported by the National Key Research and Development Program of China (grant No. 2016YFC0208503), National Natural Science Foundation of China (grant No.41605002, 41475004), Guangzhou Science and Technology Project (grant No.201604016053), Major Project of Industry-University-Research Collaborative Innovation in Guangzhou (grant No.2016201604030082) and Pearl River Nova Program of Guangzhou (grant No.201610010149). The authors gratefully acknowledge the NOAA Air Resources Laboratory (ARL) for the provision of the HYSPLIT transport and dispersion model used in this publication. We acknowledge the use of data from the NASA FIRMS application (https://firms.modaps.eosdis.nasa.gov/) operated by the NASA/Goddard Space Flight Center Earth Science Data and Information System (ESDIS) project.



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





**Table 1.** Comparisons of three $E_{abs}$ determination approaches.

| Approach | Time resolution | Temporal coverage | $E_{abs}$ determination | Instrument | Limitations |
|---|---|---|---|---|---|
| TD | minutes | months | $E_{abs} = \dfrac{\sigma_{abs\_total}}{\sigma_{abs\_pri}}$ | TD+PAS | TD temperature selection; denuded particle morphology different from emission |
| AFD | daily | years | $E_{abs} = \dfrac{\sigma_{abs\_total}}{\sigma_{abs\_pri}}$ | Filter sampler + off-line OCEC | labor intensive; only remove soluble coating |
| MAE+MRS | hourly | years | $E_{abs} = \dfrac{MAE_t}{MAE_p}$ | Aethalometer + online OCEC | MRS has minimum data points requirement, not suitable for a small dataset |





**Table 2.** Comparisons E_abs in various field studies.

| Method | Location | Sampling duration | λ nm | $E_{abs}$ | Reference |
|---|---|---|---|---|---|
| MAE | Guangzhou,China (Urban) | Jul-Sept 2017 Nov2017-Jan2018 | 520 | 1.51±0.50 1.29±0.28 | This study |
| | Guangzhou,China (Suburban) | Feb 2012-Jan 2013 | 550 | 1.50±0.48 | (Wu et al., 2018) |
| | Beijing, China (Suburban) | Nov 2014–Jan 2015 | 470 | 2.6–4.0 | (Xu et al., 2016) |
| | Beijing, China (Urban) | Nov 2014 | / | 1.66-1.91 | (Zhang et al., 2018b) |
| | Manchester, UK (Urban) | Oct2-Nov 2014 | 532 | 1.0–1.3 | (Liu et al., 2017a) |
| | Paris, France (Urban) | Mar 2014–Mar 2017 | 880 | 1.53±0.39 | (Zhang et al., 2018a) |
| | Kanpur, India (Urban) | Jan-Feb 2015 | 781 | 1.8 | (Thamban et al., 2017) |
| | Nanjing, China (Suburban) | Nov 2012 | 532 | 1.6 | (Cui et al., 2016a) |
| | Xi'an, China (Urban) | Dec 2012-Jan 2013 | 870 | 1.8 | (Wang et al., 2014) |
| TD | Shouxian, China (Rural) | Jun-Jul 2016 | 532 | 2.3 ± 0.9 | (Xu et al., 2018c) |
| | Beijing, China (Urban) | Jun 2017 | 630 | 1.59±0.26 | (Xie et al., 2019) |
| | Sacramento, USA (Urban) | Jun-Jul 2010 | 532 | 1.06± 0.01 | (Cappa et al., 2012) |
| | Fresno, USA (Urban) | Dec 2014-Jan 2015 | 532 | 1.22± 0.15 | (Cappa et al., 2019) |
| | Fontana, USA (Urban) | July 2015 | 532 | 1.07± 0.22 | |
| | Suzu, Japan | April-May 2013 | 532 | 1.06 | (Ueda et al., 2016) |
| AFD | Jinan, China (Urban) | February 2014 | 678 | 2.07± 0.72 | (Chen et al., 2017) |
| | Yuncheng, China (Rural) | Jun-Jul 2014 | 678 | 2.25± 0.55 | (Cui et al., 2016b) |





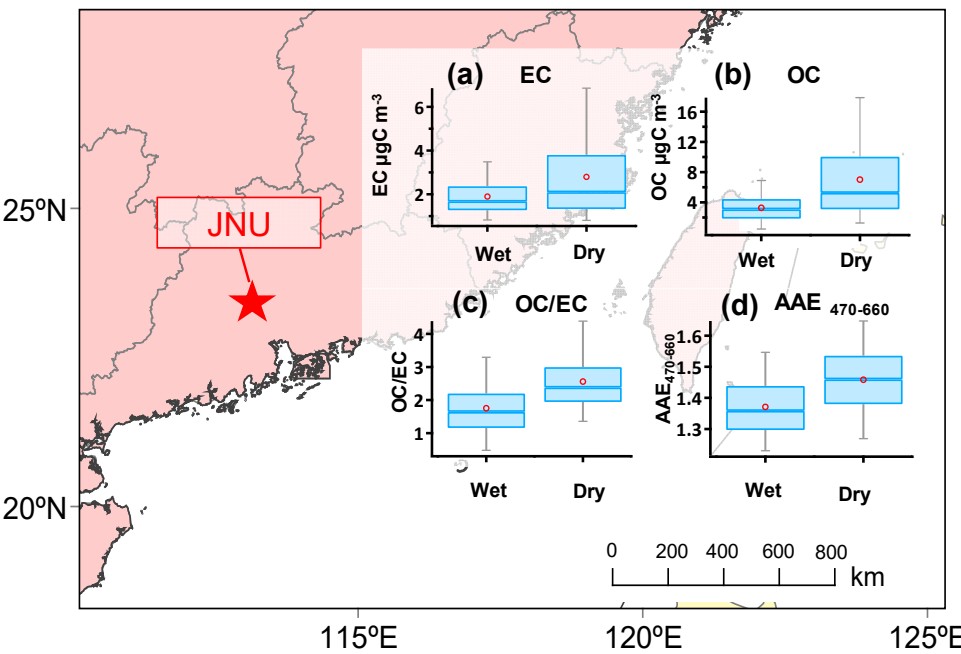

**Figure 1.** The location of the observation site. (a), (b), (c) and (d) show the box plot of EC, OC, OC/EC and AAE$_{470-660}$, respectively. Red circles represent the seasonal average. The line inside the box indicates the median. Upper and lower boundaries of the box represent the 75[th] and the 25[th] percentiles; the whiskers above and below each box represent the 95[th] and 5[th] percentiles.

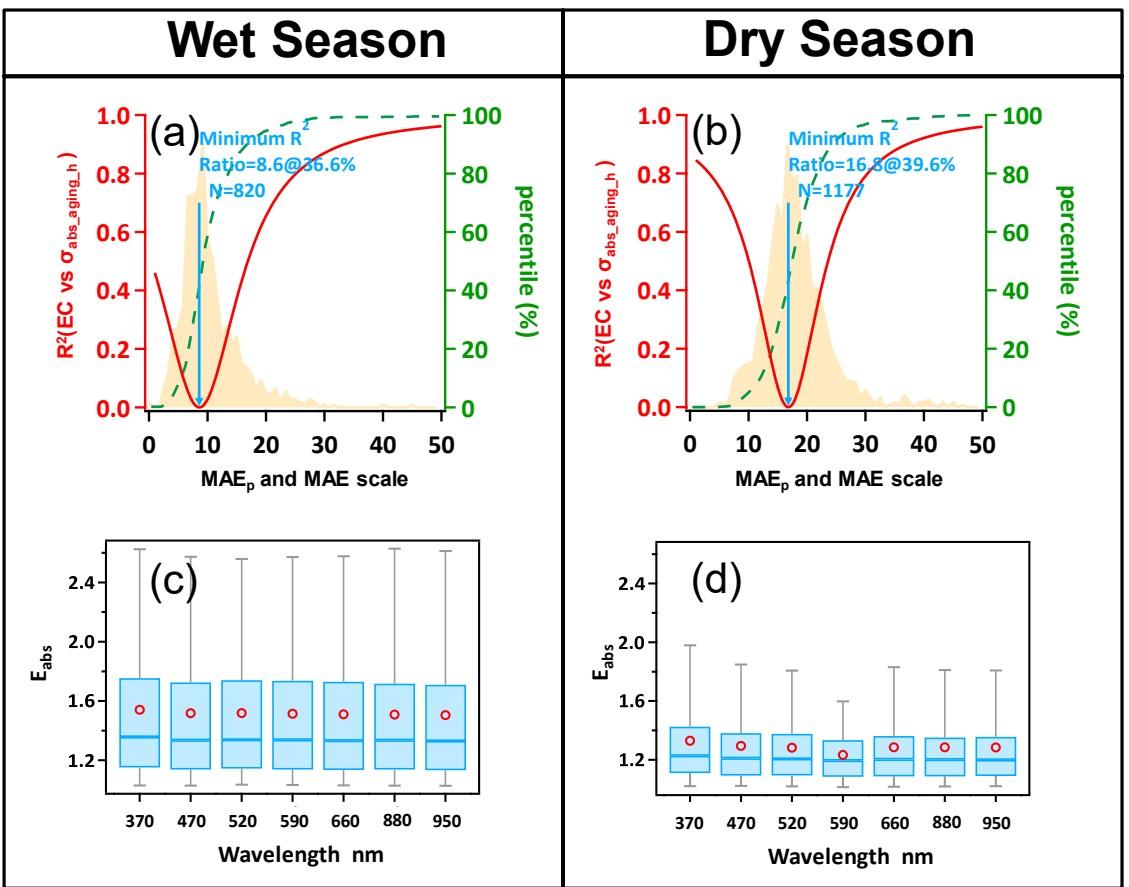

**Figure 2.** $E_{abs}$ determination by MRS. (a) Wet season $MAE_p$ determined by MRS at 520 nm. The red curve represents the correlation coefficient ($R^2$) between hypothetical $\sigma_{abs\_aging}$ ($\sigma_{abs\_total} - EC * MAE_p$) and EC mass as a function of $MAE_{p\_h}$. The shaded area in light tan represents the frequency distribution of observed MAE. The dashed green line is the cumulative distribution of observed MAE. (b) same as (a) but for dry season. (c) Spectral $E_{abs}$ determined by MRS in wet season. Red circles represent the average values. The line inside the box indicates the median. Upper and lower boundaries of the box represent the 75th and the 25th percentiles; the whiskers above and below each box represent the 95th and 5th percentiles. (d) Spectral $E_{abs}$ in dry season.





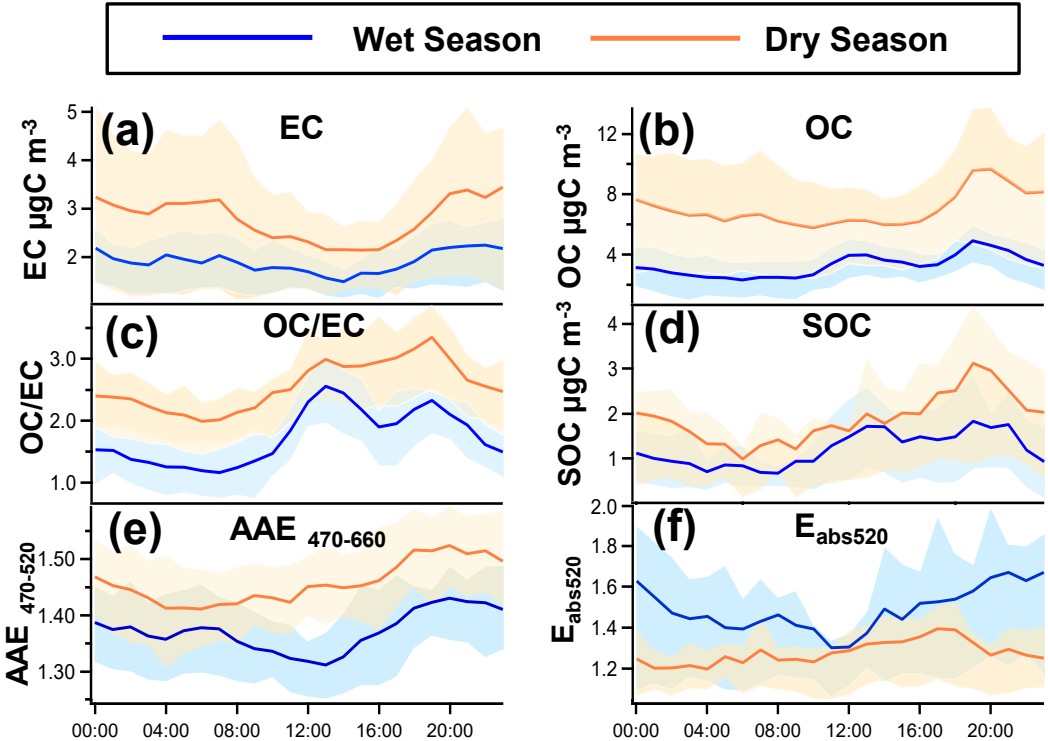

**Figure 3.** Diurnal pattern carbonaceous aerosols in wet and dry season. The solid lines represent hourly averages and the shaded areas represent $25^{th}\%$ and $75^{th}\%$ percentile. (a) EC. (b) OC. (c) OC/EC ratio. (d) SOC. (e) $AAE_{470-660}$. (f) $E_{abs520}$.

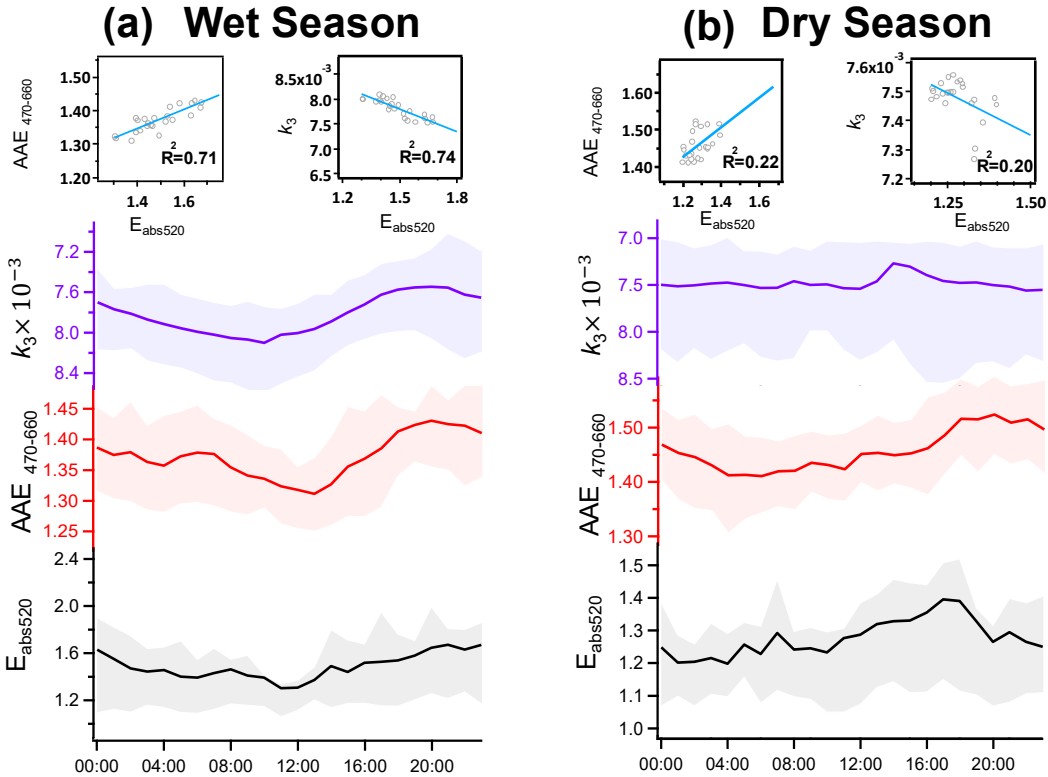

**Figure 4.** diurnal patterns of $E_{abs520}$ $AAE_{470-660}$ and $k_3$. The solid lines represent hourly averages and the shaded areas represent $25^{th}\%$ and $75^{th}\%$ percentile. (a) Wet season. (b) Dry season. It should be noted that the $k_3$ was shown on an inverted scale.





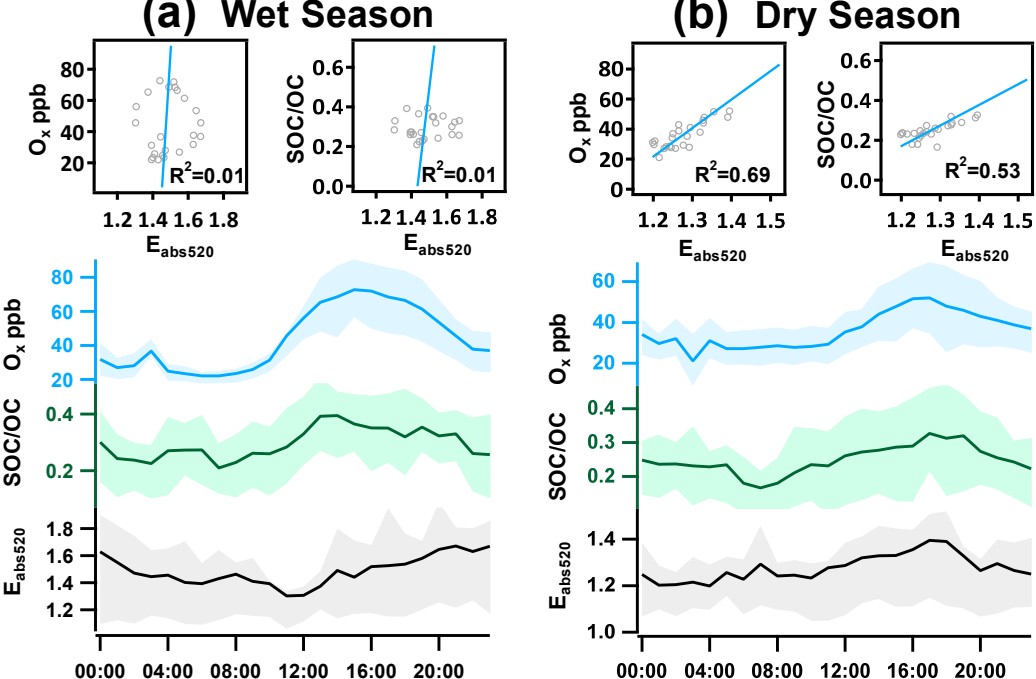

**Figure 5.** The effect of secondary process on $E_{abs520}$. (a) Diurnal pattern of $E_{abs520}$ and $O_x$ in wet season. The solid lines represent hourly averages and the shaded areas represent $25^{th}$% and $75^{th}$% percentile. (b) Same as (a) but in dry season.





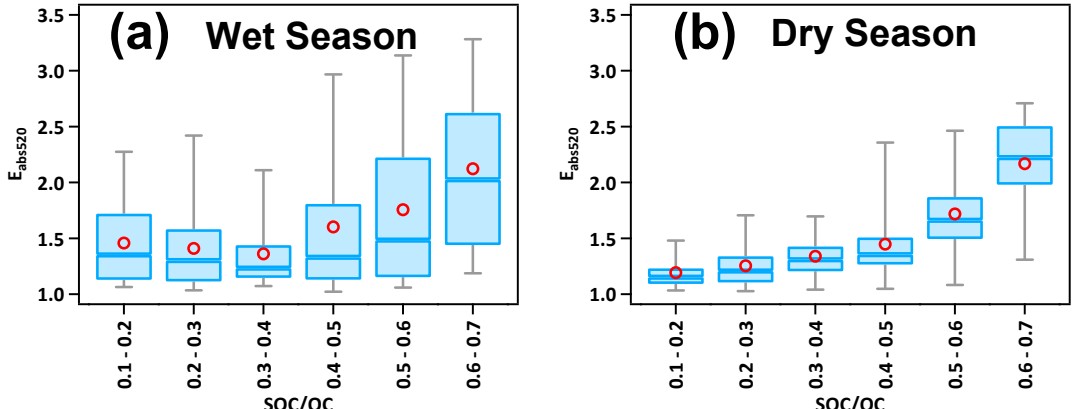

1210

**Figure 6.** $E_{abs520}$ dependency on SOC/OC ratio. (a) Wet season (b) Dry season. Red circles represent the average values. The line inside the box indicates the median. Upper and lower boundaries of the box represent the 75[th] and the 25[th] percentiles; the whiskers above and below each box represent the 95[th] and 5[th] percentiles.

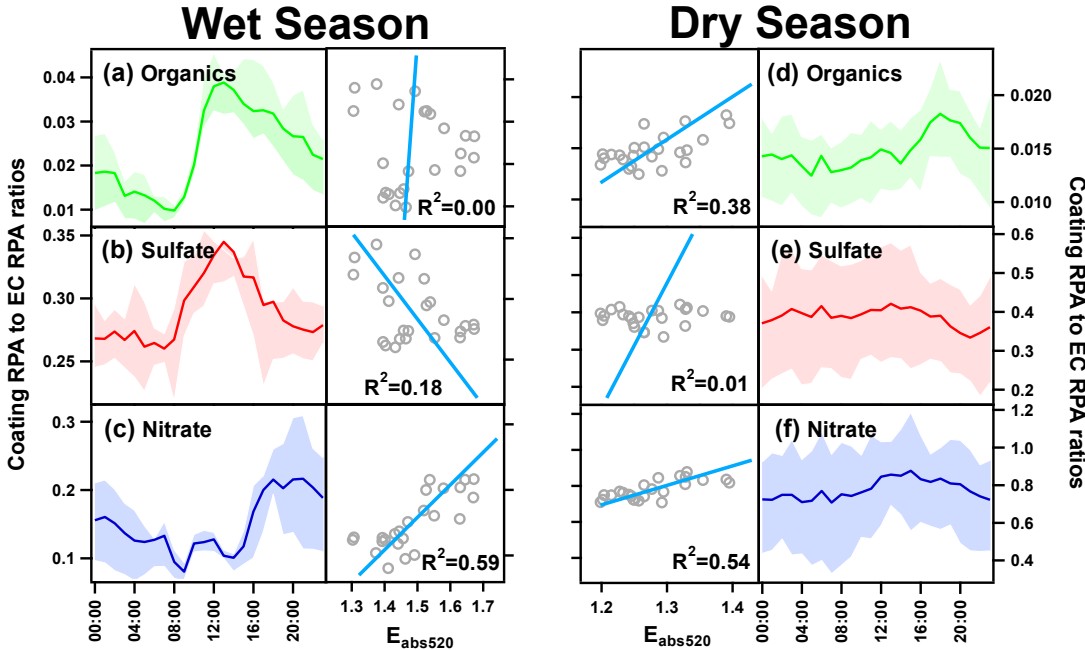

**Figure 7.** Diurnal variations of coating (including organics, sulfate and nitrate) RPA to EC RPA ratios of EC-aged particles measured by SPAMS in wet and dry seasons. The solid lines represent hourly averages and the shaded areas represent $25^{th}\%$ and $75^{th}\%$ percentile. (a) ~(c) In wet season, organics, nitrate and sulfate RPA to EC PRA ratios. The scatter plots show the corresponding correlations with $E_{abs520}$. The scatter plots share the same y axis scale with the diurnal plots. (d) ~(f) are the same as (a) ~(c) but for the dry season. Following ions are used: EC (m/z $+12[C]^+$, $+24[C_2]^+$, $+36[C_3]^+$, $+48[C_4]^+$), organics (m/z $+43[C_2H_3O]^+$), sulfate (m/z $-97 [HSO_4]^-$, $-80[SO_3]^-$) and nitrate m/z $-62 [HNO_3]^-$, $-46[NO_2]^-$ ).