# Peer review of "Amplification of black carbon light absorption induced by atmospheric aging: temporal variation at seasonal and diel scales in urban Guangzhou"

_Atmospheric Chemistry and Physics, 2019_

## Referee Comment (RC1) · Anonymous Referee #1 · 2 Dec 2019

This paper reports a field measurement about the amplification of light absorbing property of black carbon aerosol by coating using a statistical method. The results of the light property measurement from this study is interesting and quite comprehensive, which provides some valuable insights on light absorbing amplification of black carbon aerosols in highly polluted urban region in China.

Major comments: 1. The authors need to clearly state what is new in this paper and highlight which are the major new findings. I feel that the current writing style makes this paper look more like a data report of some measurement in a different location.

2. It seems that one novelty of this paper is applying a newly developed MRS method

to estimate the light absorption enhancement. If this is the key novelty, then I would recommend to add a separate section to discuss the difference between the current results from this method and those from other methods.

3. The introduction part is like a mini review rather than an introduction. It provides too many details, some of which are not quite relevant to this study. In addition, this introduction appears not to reflect the significance of raised issue/science questions.

4. When analyzing diurnal patterns of BC particles, I would always recommend to separate weekdays and weekends.

5. This paper is quite long and not well written. I think the authors should make it more concise and improve its writing.

Specific comments:

Eq3: the denominator "EC" needs to be defined first. L104: "to low" should be "too low" L108: Consider revising the sentence "Third, the TD is not the ideal time machine for reversing the morphology transformation of BC." L169: what are these "MAE values"? L269: The sentence "The MAE at the minimum R2 of the EC vs. ..." is confusing and needs to rewrite. L620: The sentence "As a result, the increasing nitrate might potentially affect BC's radiative forcing in China." should be rewritten.

---

## Referee Comment (RC2) · Anonymous Referee #2 · 11 Dec 2019

The manuscript presents a comprehensive study on the black carbon light absorption enhancement (Eabs) in urban China. They used a newly developed method for Eabs determination, which utilizes measurements from a filter-based absorption instrument and a thermal-optical analysis OC/EC analyzer. The seasonal and diurnal patterns of Eabs were analyzed, and the potential influencing factors were discussed. This manuscript includes sufficient originality, and the topic seems to fit the scope of ACP. In general, the overall quality of the manuscript is good yet the logic of some contents, especially the introduction, can be improved. I believe that the points below should be addressed. I therefore recommend a Minor Revision before publication in ACP.

[Figure]

Major comments:

1) The introduction is long but the motivation of this study seems missing. The authors should state clearly what's the scientific question that this study is trying to answer. 2) Since this study uses a new method for Eabs quantification, a comparison with previous studies should be given in more details. Table 2 provides a useful summary but corresponding discussions seems too simple in the current manuscript. 3) Figure 6 shows a clear dependency of Eabs on SOC/OC. However, the correlation between organics and Eabs is not that good as expected. The authors should explain why a good dependency was observed in Figure 6 but meanwhile a low r2 was found in Figure 7. 4) As related to comment # 3) above, one of the most interesting findings of this study is that Eabs exhibits dependency on SOC/OC ratio and has good correlation with nitrate. For Eabs dependency on SOC/OC ratio, one might believe that it is the "concentration" of SOC in total OC that affects the absorption enhancement. For good correlation between nitrate and Eabs, is there any possible reason other than partitioning behavior that would potentially contribute to the good correlation? 5) How measurement uncertainties would affect Eabs determination by MRS method? 6) The MAE values reported in this study seems higher than those reported in the literature. Any reasons? 7) The authors suggest that the correlation between Eabs and nitrate was associated with the volatility of nitrate. If that is the case, would that be applied to the organics that have a volatility similar to nitrate?

Technical comments: Line 27. "exhibit" should be "exhibited" Line 30. "were" should be "was" Line 37. "exhibit" should be "exhibited" Line 104. "to low" should be "too low" Line 592. "a Aethalometer" should be "an Aethalometer" Line 606. "two component" should be "two-component"
* * *

---

## Author Comment (AC1) · 11 Jan 2020

**Point-by-point response to review comments on manuscript acp-2019-654 "Amplification of black carbon light absorption induced by atmospheric aging: temporal variation at seasonal and diel scales in urban Guangzhou"**

**By Cheng WU on behalf of all authors**

We thank the two anonymous reviewers for their valuable time and constructive comments to improve the manuscript. Our point-by-point responses to the review comments are listed below as shown in blue. Changes to the manuscript are marked in teal blue in the revised manuscript. The marked manuscript is submitted together with this response document.

**Anonymous Referee #1**

**General comments:**

This paper reports a field measurement about the amplification of light absorbing property of black carbon aerosol by coating using a statistical method. The results of the light property measurement from this study is interesting and quite comprehensive, which provides some valuable insights on light absorbing amplification of black carbon aerosols in highly polluted urban region in China.

**Author's Response:** We appreciate the valuable time spent and efforts from the referee to improve the manuscript. Please see below for the point-by-point response to reviewers' comments.

**Major comments:**

**R1-Q1.** The authors need to clearly state what is new in this paper and highlight which are the major new findings. I feel that the current writing style makes this paper look more like a data report of some measurement in a different location.

**Author's Response:** Thanks for the suggestion. We have rewritten and reorganized many parts of the manuscript to make it more scientific-question-orientated.

For example, part of the abstract had been rewritten as follows.

Black carbon (BC) aerosols had been widely recognized as a vital climate forcer in the atmosphere. Amplification of light absorption can occur due to coatings on BC during atmospheric aging, an effect that remains uncertain in accessing the radiative forcing of

BC. Existing studies on absorption enhancement factor ($E_{abs}$) have poor coverage on both seasonal and diurnal scales. In this study, we applied a recently developed minimum R squared (MRS) method, which can cover both seasonal and diurnal scales, for $E_{abs}$ quantification. Using field measurement data in Guangzhou, the aims of this study is to explore: 1) the temporal dynamics of BC optical properties at seasonal (wet season, July 31–September 10; dry season, November 15, 2017–January 15, 2018) and diel scales (1-hour time resolution) in the typical urban environment; 2) the influencing factors on $E_{abs}$ temporal variability.

This result suggests that during the wet season, lensing effect was more likely dominating the AAE diurnal variability rather than the contribution from Brown Carbon (BrC). Secondary processing can affect $E_{abs}$ diurnal dynamics. The $E_{abs520}$ exhibited a clear dependency on secondary organic carbon to organic carbon ratio (SOC/OC), confirming the contribution of secondary organic aerosols on $E_{abs}$. $E_{abs520}$ correlated well with nitrate and showed a clear dependence on temperature. This new finding implies that gas-particle partitioning of semi-volatile compounds may potentially play an important role in steering the diurnal fluctuation of $E_{abs520}$.

**R1-Q2.** It seems that one novelty of this paper is applying a newly developed MRS method to estimate the light absorption enhancement. If this is the key novelty, then I would recommend to add a separate section to discuss the difference between the current results from this method and those from other methods.

**Author's Response:** Thanks for the suggestion. We have rewritten and reorganized abstract and introduction to emphasize the novelty of study clearly.

One rewritten example in introduction is shown below.

As summarized in Table 1, the TD approach has high time resolution but limited sampling duration, while the AFD approach potentially has long sampling duration coverage but low time resolution. As a result, $E_{abs}$ studies with both high time resolution and long sampling duration remained limited, leading to a lack of knowledge on $E_{abs}$ variability on both seasonal and diel scales. To fill this knowledge gap, the aims of this study include: 1) to explore the temporal dynamics of $E_{abs}$ on both seasonal and diel scales using the recently developed MRS approach; 2) to investigate the influencing factors on $E_{abs}$ temporal variability, including photochemical aging, biomass burning (BB) and BC mixing state. In this study, filed measurements with one-hour time resolution were conducted in urban Guangzhou, a typical megacity in southern China in both wet (July 31–September 10, 2017) and dry (November 15, 2017–January 15, 2018) seasons.

We added a new section to discuss the comparison between $E_{abs}$ by this study and previous studies.

**3.2 Comparison of $E_{abs}$ with previous studies**

Filed measurements of $E_{abs}$ values around the world are summarized in Table 2. Studies using the TD approach can achieve sub-hour time resolution, but studies using the TD approach had limited temporal coverage (normally less than a month). The AFD approach can potentially provide long-term $E_{abs}$ results as long as filter samples are

available. However, the measurement duration of existing AFD studies was less than one month as shown in Table 2. The limited temporal coverage of existing AFD studies was likely due to the intense labor involved in filter treatment. In addition, the time resolution of existing AFD studies (8–24 hour) was not sufficient to fully resolve $E_{abs}$ diurnal pattern. As a result, diurnal variations of $E_{abs}$ values at different seasons were not covered in previous studies. In comparison, the MRS approach is a good alternative to explore the $E_{abs}$ variations at both seasonal and diurnal scales. As shown in Table 2, low $E_{abs}$ values were found in California (1.06@532 nm) (Cappa et al., 2012) and Japan (1.06@532 nm) (Ueda et al., 2016). Liu et al. (2017) observed a moderate $E_{abs}$ in UK (1.0-1.3@532 nm) and suggested that the small $E_{abs}$ observed by Cappa et al. (2012) was a result of mixing state diversity. A recent study in California (Cappa et al., 2019) found moderate $E_{abs}$ at Fresno (1.22@532 nm) but low $E_{abs}$ at Fontana (1.07@532 nm), which was partially associated with unequal distribution of coating between different BC-containing particle types (Lee et al., 2019). In general, higher $E_{abs}$ values had been observed in more polluted urban areas, such as France (Paris, 1.53@880 nm) (Zhang et al., 2018a) and India (Kanpur, 1.8@781 nm) (Thamban et al., 2017). High $E_{abs}$ values had been reported in various locations in China. The $E_{abs}$ value in the wet season in our study (1.51) is higher than that in Nanjing (1.42) (Ma et al., 2019) but lower than those in central China (Shouxian, 2.3) (Xu et al., 2018), eastern China (Jinan, 1.9) (Bai et al., 2018) and northern China (Yuncheng, 2.25) (Cui et al., 2016b). The $E_{abs}$ value in the dry season in our study (1.29) is lower than those in other locations in China, such as Beijing (1.66-4.0) (Xu et al., 2016; Zhang et al., 2018b), Nanjing (1.6) (Cui et al., 2016a), Xi'an (1.8) (Wang et al., 2014) and Jinan (2.07) (Chen et al., 2017). Since the collocated comparison of the three $E_{abs}$ methods do not exist, a direct comparison between the three methods remain difficult. Nevertheless, a few studies, which conducted at the same city but during different periods, yielded comparable $E_{abs}$ values. For example, $E_{abs}$ in Nanjing by MAE method (1.6) (Cui et al., 2016a) was higher than that by the TD method (1.42) (Ma et al., 2019). This difference in $E_{abs}$ might not only due to the different $E_{abs}$ determination methods, but could also be a result of seasonal variations of $E_{abs}$.

**R1-Q3.** The introduction part is like a mini review rather than an introduction. It provides too many details, some of which are not quite relevant to this study. In addition, this introduction appears not to reflect the significance of raised issue/science questions.

**Author's Response:** Thanks for the suggestion. We agree that the introduction is longer than typical studies. Since the $E_{abs}$ determination by MRS approach is relatively new to most readers, we try to provide sufficient information to help the readers to understand the motivation of using MRS instead of conventional approaches. This background information also helps the readers to understand the working principle and advantage of MRS. Nevertheless, we agree that there are room for improving the logic flow. We have made substantial revisions in the introduction to make the manuscript more scientific-question-orientated.

**R1-Q4.** The authors need to clearly state what is new in this paper and highlight which are the major new findings. I feel that the current writing style makes this paper look more like a data report of some measurement in a different location.

**Author's Response:** Thanks for the suggestion. We rewritten some of the contents to show the new findings clearly.

An example in abstract is shown below.

$E_{abs520}$ correlated well with nitrate and show a clear dependence on temperature. This new finding implies that gas-particle partitioning of semi-volatile compounds may potentially play an important role in steering the diurnal fluctuation of $E_{abs520}$.

**R1-Q5.** When analyzing diurnal patterns of BC particles, I would always recommend to separate weekdays and weekends.

**Author's Response:** Thanks for the suggestion. Now we included Figure S10 to show the weekday/weekend effect on diurnal patterns. Corresponding discussions were added and also shown below.

To explore the effect of traffic volume, the weekday/weekend effect was investigated in Figure S10. The evening EC peak reduced substantially during weekend, implying that traffic volume has a strong influence on shaping the diurnal pattern of EC.

The difference of diurnal OC/EC pattern between weekday and weekend is negligible (Figure S10), suggesting that the portion of different vehicle type (e.g. diesel vs. gasoline) is relatively constant between weekday and weekend.

**R1-Q6.** This paper is quite long and not well written. I think the authors should make it more concise and improve its writing.

**Author's Response:** Thanks for the suggestion. To shorten the length of the manuscript, we had moved back trajectory and wind rose discussions to the supplement. We had also made necessary revisions to improve the conciseness of the manuscript.

**Specific comments:**

**R1-Q7.** Eq3: the denominator "EC" needs to be defined first.

**Author's Response:** Thanks for pointing out. The following content is added.

EC in Eq. 3 and 4 represents elemental carbon mass concentration determined by the thermal optical analysis method (Wu et al., 2012), which can be considered as a surrogate of BC mass concentration.

**R1-Q8.** L104: "to low" should be "too low"

**Author's Response:** Thanks for pointing out the typo. Revision made.

**R1-Q9.** L108: Consider revising the sentence "Third, the TD is not the ideal time machine for reversing the morphology transformation of BC."

**Author's Response:** Thanks for the suggestion. The sentence had been rewritten as shown below.

Third, the TD approach cannot perfectly reverse the morphology transformation of BC from aged state back to freshly emitted state.

**R1-Q10.** L169: what are these "MAE values"?

**Author's Response:** Thanks for the suggestion. We add Table S1 to show theses literature values and the sentence had been rewritten as shown below.

MAE values from study by Drinovec et al. (2015) was adopted for $\sigma_{abs\_total}$ back-calculations from eBC at different wavelengths as shown in Table S1.

**Table S1.** MAE values from study by Drinovec et al. (2015) was adopted for $\sigma_{abs\_total}$ back-calculations at different wavelengths.

| Wavelength (nm) | MAE ($m^2g^{-1}$) |
|---|---|
| 370 | 18.47 |
| 470 | 14.54 |
| 520 | 13.14 |
| 590 | 11.58 |
| 660 | 10.35 |
| 880 | 7.77 |
| 950 | 7.19 |

**R1-Q11.** L269: The sentence "The MAE at the minimum R2 of the EC vs. …" is confusing and needs to rewrite.

**Author's Response:** Thanks for the suggestion. The sentence had been rewritten as shown below.

In MRS calculation, the correlation ($R^2$) between measured EC and estimated hypothetical $\sigma_{abs\_aging}$ is examined as a function of a series of hypothetical $MAE_p$ ($MAE_{p\_h}$). Since $\sigma_{abs\_aging}$ was resulted from secondary processing while EC was coming from primary emissions, a $MAE_{p\_h}$ that leads to a minimum $R^2$(EC, $\sigma_{abs\_aging\_h}$) can best represents the independent nature between EC and $\sigma_{abs\_aging}$. As a result, $MAE_{p\_h}$ at minimum $R^2$(EC, $\sigma_{abs\_aging\_h}$) corresponds to the authentic $MAE_p$.

**R1-Q12.** L620: The sentence "As a result, the increasing nitrate might potentially affect BC's radiative forcing in China." should be rewritten.

**Author's Response:** Thanks for the suggestion. The sentence had been rewritten as shown below.

If the nitrate fraction in the coating materials on BC increases, the diurnal pattern of $E_{abs}$ for BC may be affected by the fluctuation of nitrate content in aerosol particles. As a result, the increasing concentration of nitrate might potentially affect radiative forcing by BC in China.

**Anonymous Referee #2**

**General comments:**

The manuscript presents a comprehensive study on the black carbon light absorption enhancement (Eabs) in urban China. They used a newly developed method for Eabs determination, which utilizes measurements from a filter-based absorption instrument and a thermal-optical analysis OC/EC analyzer. The seasonal and diurnal patterns of Eabs were analyzed, and the potential influencing factors were discussed. This manuscript includes sufficient originality, and the topic seems to fit the scope of ACP. In general, the overall quality of the manuscript is good yet the logic of some contents, especially the introduction, can be improved. I believe that the points below should be addressed. I therefore recommend a Minor Revision before publication in ACP.

**Author's Response:** We appreciate the valuable time spent and efforts from the referee to improve the manuscript. Please see below for point-by-point response to reviewers' comments.

**Major comments:**

**R2-Q1.** The introduction is long but the motivation of this study seems missing. The authors should state clearly what's the scientific question that this study is trying to answer.

**Author's Response:** Thanks for the suggestion. We have made substantial revisions in the introduction section by emphasizing the motivation and scientific question.

The motivation of this study is emphasized as shown below.

As summarized in Table 1, the TD approach has high time resolution but limited sampling duration, while the AFD approach potentially has long sampling duration coverage but low time resolution. As a result, $E_{abs}$ studies with both high time resolution and long sampling duration remained limited, leading to a lack of knowledge on $E_{abs}$ variability on both seasonal and diel scales. To fill this knowledge gap, the aims of this study include: 1) to explore the temporal dynamics of $E_{abs}$ on both seasonal and diel scales using the recently developed MRS approach; 2) to investigate the influencing factors on $E_{abs}$ temporal variability, including photochemical aging, biomass burning (BB) and BC mixing state. In this study, filed measurements with one-hour time resolution were conducted in urban Guangzhou, a typical megacity in southern China in both wet (July 31–September 10) and dry (November 15, 2017–January 15, 2018) seasons.

**R2-Q2.** Since this study uses a new method for Eabs quantification, a comparison with previous studies should be given in more details. Table 2 provides a useful summary but corresponding discussions seems too simple in the current manuscript.

**Author's Response:** Thanks for the suggestion. We added a separated section to discuss the comparisons of the current study and previous studies.

**3.2 Comparison of $E_{abs}$ with previous studies**

Filed measurements of $E_{abs}$ values around the world are summarized in Table 2. Studies using the TD approach can achieve sub-hour time resolution, but studies using the TD approach had limited temporal coverage (normally less than a month). The AFD approach can potentially provide long-term $E_{abs}$ results as long as filter samples are available. However, the measurement duration of existing AFD studies was less than one month as shown in Table 2. The limited temporal coverage of existing AFD studies was likely due to the intense labor involved in filter treatment. In addition, the time resolution of existing AFD studies (8–24 hour) was not sufficient to fully resolve $E_{abs}$ diurnal pattern. As a result, diurnal variations of $E_{abs}$ values at different seasons were not covered in previous studies. In comparison, the MRS approach is a good alternative to explore the $E_{abs}$ variations at both seasonal and diurnal scales. As shown in Table 2, low $E_{abs}$ values were found in California (1.06@532 nm) (Cappa et al., 2012) and Japan (1.06@532 nm) (Ueda et al., 2016). Liu et al. (2017) observed a moderate $E_{abs}$ in UK (1.0-1.3@532 nm) and suggested that the small $E_{abs}$ observed by Cappa et al. (2012) was a result of mixing state diversity. A recent study in California (Cappa et al., 2019) found moderate $E_{abs}$ at Fresno (1.22@532 nm) but low $E_{abs}$ at Fontana (1.07@532 nm), which was partially associated with unequal distribution of coating between different BC-containing particle types (Lee et al., 2019). In general, higher $E_{abs}$ values had been observed in more polluted urban areas, such as France (Paris, 1.53@880 nm) (Zhang et al., 2018a) and India (Kanpur, 1.8@781 nm) (Thamban et al., 2017). High $E_{abs}$ values had been reported in various locations in China. The $E_{abs}$ value in the wet season in our study (1.51) is higher than that in Nanjing (1.42) (Ma et al., 2019) but lower than those in central China (Shouxian, 2.3) (Xu et al., 2018), eastern China (Jinan, 1.9) (Bai et al., 2018) and northern China (Yuncheng, 2.25) (Cui et al., 2016b). The $E_{abs}$ value in the dry season in our study (1.29) is lower than those in other locations in China, such as Beijing (1.66-4.0) (Xu et al., 2016; Zhang et al., 2018b), Nanjing (1.6) (Cui et al., 2016a), Xi'an (1.8) (Wang et al., 2014) and Jinan (2.07) (Chen et al., 2017). Since the collocated comparison of the three $E_{abs}$ methods do not exist, a direct comparison between the three methods remain difficult. Nevertheless, a few studies, which conducted at the same city but during different periods, yielded comparable $E_{abs}$ values. For example, $E_{abs}$ in Nanjing by MAE method (1.6) (Cui et al., 2016a) was higher than that by the TD method (1.42) (Ma et al., 2019). This difference in $E_{abs}$ might not only due to the different $E_{abs}$ determination methods, but could also be a result of seasonal variations of $E_{abs}$.

**R2-Q3.** Figure 6 shows a clear dependency of $E_{abs}$ on SOC/OC. However, the correlation between organics and $E_{abs}$ is not that good as expected. The authors should explain why a good dependency was observed in Figure 6 but meanwhile a low $r^2$ was found in Figure 7.

**Author's Response:** Thanks for the comments. There are several possibilities for the lower $R^2$ of organics in Figure 7. First, Organics shown in Figure 7 contain both primary and secondary organics, two types or organics that might have different impacts on $E_{abs}$ values. Second, the dependency of $E_{abs}$ on SOC/OC might not necessarily be reflected in the form of correlation on a diurnal scale.

The following contents in section 3.6 were rephased.

It should be noted that a good $E_{abs}$ dependence on SOC/OC observed in Figure 6 does not necessarily lead to a good diurnal correlation between $E_{abs}$ and SOC/OC (e.g. Figure 4a). In other words, the dependency of $E_{abs}$ on SOC/OC might not necessarily be reflected in the form of correlation on a diurnal scale. Thus, the poor diurnal correlation between SOC/OC and $E_{abs520}$ observed in the wet season (Figure 4a) cannot rule out the contribution of SOC on $E_{abs520}$.

The following contents in section 3.7 were rephased.

There are two possibilities for the lower $R^2$ (Organics, $E_{abs}$) in Figure 7d comparing to $R^2$(SOC/OC, $E_{abs}$) in Figure 4b. First, organics shown in Figure 7d contain both primary and secondary organics, while SOC/OC shown in Figure 4b represents the secondary portion only. Second, poor diurnal correlations do not necessarily rule out the contribution of organics and sulfate to $E_{abs}$ by analogy with the SOC correlation with $E_{abs}$ as discussed in Section 3.6.

**R2-Q4.** As related to comment # 3) above, one of the most interesting findings of this study is that Eabs exhibits dependency on SOC/OC ratio and has good correlation with nitrate. For Eabs dependency on SOC/OC ratio, one might believe that it is the "concentration" of SOC in total OC that affects the absorption enhancement. For good correlation between nitrate and Eabs, is there any possible reason other than partitioning behavior that would potentially contribute to the good correlation?

**Author's Response:** Thanks for the suggestion. Besides lensing effect of non-absorbing coatings (e.g. nitrate) induced light absorption enhancement, formation of light-absorbing components could also lead to light absorption enhancement. For example, nitro-aromatic compounds (NACs) are secondary formed light-absorbing components that contribute light absorption enhancement. Since NAC formation is strongly associated with $NO_x$, a good correlation between NACs and nitrate had been observed (Chow et al., 2016). However, NACs are part of Brown Carbon (BrC), thus have strong wavelength dependency. If a considerable fraction of NACs is present in $PM_{2.5}$, an elevated AAE would be observed. The AAE observed in this study was <2, implying a weak signature of BrC. In our study, the abundance of NACs was not measured, therefore $E_{abs}$ contribution from NACs remains unclear. But considering the fact that typical NAC concentration in $PM_{2.5}$ was low (<10 ng m$^{-3}$) (Chow et al., 2016) and their contribution to light absorption was small (<1%@370 nm) (Teich et al., 2017), the contribution of NACs to $R^2$ (Nitrate, $E_{abs}$) is expected to be very small.

**R2-Q5.** How measurement uncertainties would affect $E_{abs}$ determination by MRS method?

**Author's Response:** The measurement uncertainties of MRS method for $E_{abs}$ determination had been systematically examined in our previous study (Wu et al., 2018). Thus, only a brief introduction is given here. Three scenarios are considered using the data from our previous study (Wu et al., 2018). Scenario A represents systematic biases in Aehtalometer and OCEC measurements in the form of multipliers. The biased data are marked as $\sigma'_{abs550}$ and EC', respectively, as shown below:

$$\sigma'_{abs550} = \sigma_{abs550} \times 2 \qquad \text{(R-1)}$$
$$EC' = EC \times 0.7 \qquad \text{(R-2)}$$

The $E_{abs}$ remain the same in scenario A, suggesting that MRS is not sensitive to the systematic biases that are in the form of multipliers.

In scenario B, EC by different TOA protocols are compared to investigate the effect of different EC determination approaches while $\sigma_{abs550}$ remains unchanged. EC by IMPROVE TOR protocol is calculated from NIOSH TOT EC following an empirical formula for suburban sites derived from a 3-year OCEC dataset in PRD (Wu et al., 2016):

$$EC_{IMP\_TOR} = 2.63 \times EC_{NSH\_TOT} + 0.05 \qquad \text{(R-3)}$$

In scenario B, EC uncertainty induced $E_{abs}$ change is small (from 1.44 to 1.40) as shown in Figure R2, suggesting that the MRS is not sensitive to the systematic biases that contain both multipliers and offsets.

Scenario C examines the impact of sample-dependent bias as a function of $E_{abs}$. Unlike the proportional bias in Scenario A and B that is the same for all data points, the bias in Scenario C depends on the $E_{abs550}$ of individual samples, which are parametrized by Eqs. (R-4) and (R-5).

$$\sigma'_{abs550} = \sigma_{abs550} + \sigma_{abs550} \times (k \times E_{abs550} - k) \qquad \text{(R-4)}$$

$$EC' = EC - EC \times (k \times E_{abs550} - k) \qquad \text{(R-5)}$$

As shown in Eqs. (R-4) and (R-5), the positive bias of $\sigma_{abs550}$ and negative bias of EC are proportional to $E_{abs550}$. The magnitude of $E_{abs550}$-dependent bias is regulated by the factor k. Since $\sigma'_{abs550}$ and EC' are biased in different directions, resulting a further amplification in MAE biases, which could be considered as the extreme case. As shown in Figure R3, for k=10% (corresponding to a bias of 10% when $E_{abs}$=2), the bias of MRS-derived $E_{abs}$ is very small (1%). For k=20%, the MRS-derived $E_{abs}$ changes from 1.44 to 1.66, leading to a bias of 15%. These results imply that if the measurement bias follows the same form as demonstrated in scenario C, the bias is not negligible but still acceptable. If the impact only affects $\sigma_{abs}$ or EC rather than impacting both, the bias is expected to be smaller than the estimation shown in Scenario C.

In summary, $E_{abs}$ determination by MRS method is not sensitive to measurement uncertainties in light absorption or EC mass.

[Figure]

**Figure R1.** Comparison of $E_{abs}$ from original data and systematically biased data (Scenario A). It should be noted that the $E_{abs}$ shown here is ratio of averages, which is different form the annual average $E_{abs}$ calculated from average of ratios.

[Figure]

**Figure R2.** Comparison of $E_{abs}$ from data using NIOSH EC and data using IMPROVE EC (Scenario B). It should be noted that the $E_{abs}$ shown here is ratio of averages, which is different form the annual average $E_{abs}$ calculated from average of ratios.

[Figure]

**Figure R3.** Comparison of $E_{abs}$ from data using original data and $E_{abs}$ depended biased data (Scenario C). It should be noted that the $E_{abs}$ shown here is ratio of averages, which is different form the annual average $E_{abs}$ calculated from average of ratios.

**R2-Q6.** The MAE values reported in this study seems higher than those reported in the literature. Any reasons?

**Author's Response:** Two possible factors can lead to high MAE values. The first possibility is associated with measurement uncertainty of MAE, which can be further attributed to two uncertainties: $\sigma_{abs}$ determination uncertainty (AE33) and EC determination uncertainty (OCEC analyzer). Both overestimation of $\sigma_{abs}$ or underestimation of EC can lead to overestimation of MAE. Over estimation of $\sigma_{abs}$ could happen when the AE33 correction factor $C_{ref}$ is underestimated, since $C_{ref}$ could vary by locations and changed over time. EC determination could be different by a factor of 2-4 due to different analysis protocols (Chow et al., 2004; Wu et al., 2012). The second possibility is that the particles measured in this study were highly aged, leading to higher MAE values than those in previous studies. Nevertheless, the focus of this study is $E_{abs}$, whose determination procedure by MRS is not sensitive to measurement biases in MAE, as discussed in response to R2-Q5.

**R2-Q7.** The authors suggest that the correlation between Eabs and nitrate was associated with the volatility of nitrate. If that is the case, would that be applied to the organics that have a volatility similar to nitrate?

**Author's Response:** We agreed with the reviewer that organics with a volatility similar to nitrate could potentially contribute to the diurnal variability of $E_{abs}$. This possibility is mentioned in section 3.7.

By analogy with nitrate, organic compounds with a volatility similar to nitrate might potentially involve in shaping the diurnal pattern of $E_{abs}$ in the wet season.

**Technical comments:**

**R2-Q8.** Line 27. "exhibit" should be "exhibited"

**R2-Q9.** Line 30. "were" should be "was"

**R2-Q10.** Line 37. "exhibit" should be "exhibited"

**R2-Q11.** Line 104. "to low" should be "too low"

**R2-Q12.** Line 592. "a Aethalometer" should be "an Aethalometer"

**R2-Q13.** Line 606. "two component" should be "two-component"

**Author's Response:** Thanks for the technical suggestions. Revisions made accordingly.

**References**

Bai, Z., Cui, X., Wang, X., Xie, H., and Chen, B.: Light absorption of black carbon is doubled at Mt. Tai and typical urban area in North China, Sci.Total.Environ., 635, 1144-1151, doi: 10.1016/j.scitotenv.2018.04.244, 2018.

Cappa, C. D., Onasch, T. B., Massoli, P., Worsnop, D. R., Bates, T. S., Cross, E. S., Davidovits, P., Hakala, J., Hayden, K. L., Jobson, B. T., Kolesar, K. R., Lack, D. A., Lerner, B. M., Li, S.-M., Mellon, D., Nuaaman, I., Olfert, J. S., Petäjä, T., Quinn, P. K., Song, C., Subramanian, R., Williams, E. J., and Zaveri, R. A.: Radiative Absorption Enhancements Due to the Mixing State of Atmospheric Black Carbon, Science, 337, 1078-1081, doi: 10.1126/science.1223447, 2012.

Cappa, C. D., Zhang, X., Russell, L. M., Collier, S., Lee, A. K. Y., Chen, C.-L., Betha, R., Chen, S., Liu, J., Price, D. J., Sanchez, K. J., McMeeking, G. R., Williams, L. R., Onasch, T. B., Worsnop, D. R., Abbatt, J., and Zhang, Q.: Light absorption by ambient black and brown carbon and its dependence on black carbon coating state for two California, USA cities in winter and summer, J. Geophys. Res., 124, 1550-1577, doi: 10.1029/2018JD029501, 2019.

Chen, B., Bai, Z., Cui, X., Chen, J., Andersson, A., and Gustafsson, Ö.: Light absorption enhancement of black carbon from urban haze in Northern China winter, Environ. Pollut., 221, 418-426, doi: 10.1016/j.envpol.2016.12.004, 2017.

Chow, J. C., Watson, J. G., Chen, L. W. A., Arnott, W. P., and Moosmuller, H.: Equivalence of elemental carbon by thermal/optical reflectance and transmittance with different temperature protocols, Environ. Sci. Technol., 38, 4414-4422, doi: 10.1021/Es034936u, 2004.

Chow, K. S., Huang, X. H. H., and Yu, J. Z.: Quantification of nitroaromatic compounds in atmospheric fine particulate matter in Hong Kong over 3 years: field measurement evidence for secondary formation derived from biomass burning emissions, Environmental Chemistry, 13, 665-673, doi: 10.1071/EN15174, 2016.

Cui, F., Chen, M., Ma, Y., Zheng, J., Zhou, Y., Li, S., Qi, L., and Wang, L.: An intensive study on aerosol optical properties and affecting factors in Nanjing, China, Journal of Environmental Sciences, 40, 35-43, doi: 10.1016/j.jes.2015.08.017, 2016a.

Cui, X., Wang, X., Yang, L., Chen, B., Chen, J., Andersson, A., and Gustafsson, Ö.: Radiative absorption enhancement from coatings on black carbon aerosols, Sci.Total.Environ., 551, 51-56, doi: 10.1016/j.scitotenv.2016.02.026, 2016b.

Drinovec, L., Močnik, G., Zotter, P., Prévôt, A. S. H., Ruckstuhl, C., Coz, E., Rupakheti, M., Sciare, J., Müller, T., Wiedensohler, A., and Hansen, A. D. A.: The "dual-spot" Aethalometer: an improved measurement of aerosol black carbon with real-time loading compensation, Atmos. Meas. Tech., 8, 1965-1979, doi: 10.5194/amt-8-1965-2015, 2015.

Lee, A. K. Y., Rivellini, L.-H., Chen, C.-L., Liu, J., Price, D., Betha, R., Russell, L. M., Zhang, X., and Cappa, C. D.: Influences of primary emission and secondary coating formation on the particle diversity and mixing state of black carbon particles, Environ. Sci. Technol., 53, 9429-9438, doi: 10.1021/acs.est.9b03064, 2019.

Liu, D., Whitehead, J., Alfarra, M. R., Reyes-Villegas, E., Spracklen, D. V., Reddington, C. L., Kong, S., Williams, P. I., Ting, Y.-C., Haslett, S., Taylor, J. W., Flynn, M. J., Morgan, W. T., McFiggans, G., Coe, H., and Allan, J. D.: Black-carbon absorption enhancement in the atmosphere determined by particle mixing state, Nature Geosci, 10, 184-188, doi: 10.1038/ngeo2901, 2017.

Ma, Y., Huang, C., Jabbour, H., Zheng, Z., Wang, Y., Jiang, Y., Zhu, W., and Zheng, J.: Mixing state and light absorption enhancement of black carbon aerosols in summertime Nanjing, China, Atmos. Environ., 117141, doi: 10.1016/j.atmosenv.2019.117141, 2019.

Teich, M., van Pinxteren, D., Wang, M., Kecorius, S., Wang, Z., Müller, T., Močnik, G., and Herrmann, H.: Contributions of nitrated aromatic compounds to the light absorption of water-soluble and particulate brown carbon in different atmospheric environments in Germany and China, Atmos. Chem. Phys., 17, 1653-1672, doi: 10.5194/acp-17-1653-2017, 2017.

Thamban, N. M., Tripathi, S. N., Moosakutty, S. P., Kuntamukkala, P., and Kanawade, V. P.: Internally mixed black carbon in the Indo-Gangetic Plain and its effect on absorption enhancement, Atmos Res, 197, 211-223, doi: 10.1016/j.atmosres.2017.07.007, 2017.

Ueda, S., Nakayama, T., Taketani, F., Adachi, K., Matsuki, A., Iwamoto, Y., Sadanaga, Y., and Matsumi, Y.: Light absorption and morphological properties of soot-containing aerosols observed at an East Asian outflow site, Noto Peninsula, Japan, Atmos. Chem. Phys., 16, 2525-2541, doi: 10.5194/acp-16-2525-2016, 2016.

Wang, Q. Y., Huang, R. J., Cao, J. J., Han, Y. M., Wang, G. H., Li, G. H., Wang, Y. C., Dai, W. T., Zhang, R. J., and Zhou, Y. Q.: Mixing State of Black Carbon Aerosol in a Heavily Polluted Urban Area of China: Implications for Light Absorption Enhancement, Aerosol. Sci. Technol., 48, 689-697, doi: 10.1080/02786826.2014.917758, 2014.

Wu, C., Ng, W. M., Huang, J., Wu, D., and Yu, J. Z.: Determination of Elemental and Organic Carbon in PM2.5 in the Pearl River Delta Region: Inter-Instrument (Sunset vs. DRI Model 2001 Thermal/Optical Carbon Analyzer) and Inter-Protocol Comparisons (IMPROVE vs. ACE-Asia Protocol), Aerosol. Sci. Technol., 46, 610-621, doi: 10.1080/02786826.2011.649313, 2012.

Wu, C., Huang, X. H. H., Ng, W. M., Griffith, S. M., and Yu, J. Z.: Inter-comparison of NIOSH and IMPROVE protocols for OC and EC determination: implications for inter-protocol data conversion, Atmos. Meas. Tech., 9, 4547-4560, doi: 10.5194/amt-9-4547-2016, 2016.

Wu, C., Wu, D., and Yu, J. Z.: Quantifying black carbon light absorption enhancement with a novel statistical approach, Atmos. Chem. Phys., 18, 289-309, doi: 10.5194/acp-18-289-2018, 2018.

Xu, X., Zhao, W., Zhang, Q., Wang, S., Fang, B., Chen, W., Venables, D. S., Wang, X., Pu, W., Wang, X., Gao, X., and Zhang, W.: Optical properties of atmospheric fine particles near Beijing during the HOPE-J3A campaign, Atmos. Chem. Phys., 16, 6421-6439, doi: 10.5194/acp-16-6421-2016, 2016.

Xu, X., Zhao, W., Qian, X., Wang, S., Fang, B., Zhang, Q., Zhang, W., Venables, D. S., Chen, W., Huang, Y., Deng, X., Wu, B., Lin, X., Zhao, S., and Tong, Y.: Influence of photochemical aging on light absorption of atmospheric black carbon and aerosol single scattering albedo, Atmos. Chem. Phys., 18, 16829-16844, doi: 10.5194/acp-2018-59, 2018.

Zhang, Y., Favez, O., Canonaco, F., Liu, D., Močnik, G., Amodeo, T., Sciare, J., Prévôt, A. S. H., Gros, V., and Albinet, A.: Evidence of major secondary organic aerosol contribution to lensing effect black carbon absorption enhancement, npj Climate and Atmospheric Science, 1, 47, doi: 10.1038/s41612-018-0056-2, 2018a.

Zhang, Y., Zhang, Q., Cheng, Y., Su, H., Li, H., Li, M., Zhang, X., Ding, A., and He, K.: Amplification of light absorption of black carbon associated with air pollution, Atmos. Chem. Phys., 18, 9879-9896, doi: 10.5194/acp-18-9879-2018, 2018b.